# ZIP: An Efficient Zeroth-order Prompt Tuning for Black-box Vision-Language Models

**Seonghwan Park[1], Jaehyeon Jeong[1], Yongjun Kim[1], Jaeho Lee[1,2], Namhoon Lee[1,2]**
[1]POSTECH, [2]Yonsei University
{seonghwan.park,chungsml,kus0505,jaeho.lee,namhoonlee}@postech.ac.kr

## Abstract

Recent studies have introduced various approaches for prompt-tuning black-box vision-language models, referred to as black-box prompt-tuning (BBPT). While BBPT has demonstrated considerable potential, it is often found that many existing methods require an excessive number of queries (*i.e.*, function evaluations), which poses a significant challenge in real-world scenarios where the number of allowed queries is limited. To tackle this issue, we propose Zeroth-order Intrinsic-dimensional Prompt-tuning (ZIP), a novel approach that enables efficient and robust prompt optimization in a purely black-box setting. The key idea of ZIP is to reduce the problem dimensionality and the variance of zeroth-order gradient estimates, such that the training is done fast with far less queries. We achieve this by re-parameterizing prompts in low-rank representations and designing intrinsic-dimensional clipping of estimated gradients. We evaluate ZIP on 13+ vision-language tasks in standard benchmarks and show that it achieves an average improvement of approximately 6% in few-shot accuracy and 48% in query efficiency compared to the best-performing alternative BBPT methods, establishing a new state of the art. Our ablation analysis further shows that the proposed clipping mechanism is robust and nearly optimal, without the need to manually select the clipping threshold, matching the result of expensive hyperparameter search.

## 1 Introduction

Foundation models pre-trained on a vast amount of data are creating tremendous success across a wide range of applications in various domains (Ramesh et al., 2021; Radford et al., 2021; Jia et al., 2021; Singh et al., 2022; Liu et al., 2023a; Copet et al., 2024). A notable example is CLIP (Radford et al., 2021) which learns visual concepts from natural language supervision and works zero-shot at inference.

In fact, these models are fine-tuned for specific downstream tasks at deployment to create yet more performance refinement in practice (Liu et al., 2022). It is noteworthy, however, that fine-tuning these models is not only computationally expensive, but also requires full access to model specifications. The complication here is that many high-performing foundation models are provided only as a software-as-a-service (OpenAI, 2023; Google, 2023) without model details due to commercial interests and security concerns.

To overcome this challenge, recent works have suggested to fine-tune such *black-box* models via so-called black-box prompt-tuning (BBPT) (Sun et al., 2022b; Diao et al., 2023; Oh et al., 2023; Yu et al., 2023); *i.e.*, by parameterizing input prompts and only optimizing them via classic derivative-free optimization methods such as evolutionary strategies (Hansen & Ostermeier, 2001; Hansen et al., 2003) and zeroth-order optimization (Spall, 1992; 1997; Ghadimi & Lan, 2013), it enables fine-tuning without direct access to the model details or back-propagation.

Nevertheless, it still remains a major challenge to secure query-efficiency in existing BBPT methods. Specifically, we find that many existing approaches require excessive model evaluations (*i.e.*, queries), often spanning several tens of thousands times (Tsai et al., 2020; Oh et al., 2023), and moreover, they result in significant performance drop when they are given a limited query budget. This is quite critical in many practical scenarios where large models are provided in the form of prediction APIs, and users can only make use of it with a limited budget.

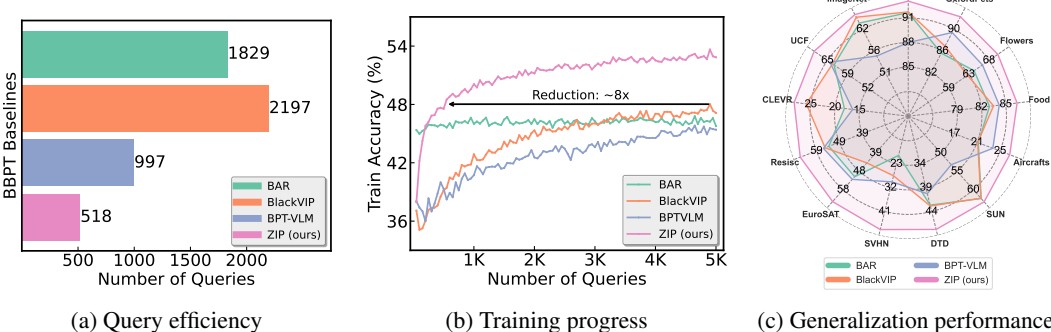

(a) Query efficiency       (b) Training progress       (c) Generalization performance

Figure 1: ZIP performance summary: (a) number of queries required to reach a given target accuracy, (b) training accuracy vs. number of queries, and (c) test set accuracies, measured on 13 standard vision-language tasks; the first two are arithmetic mean. ZIP shows its outstanding performance compared to other black-box prompt-tuning methods (*e.g.*, BAR (Tsai et al., 2020), BLACKVIP (Oh et al., 2023), BPTVLM (Yu et al., 2023); the details are provided in Appendix C.5) in terms of both training and generalization performances.

In this work, we propose a new method called ZIP: Zeroth-order Intrinsic-dimensional Prompt-tuning to tackle this challenge. The key idea is to reduce the dimensionality of the problem (hence the term "intrinsic") and the variance of zeroth-order gradients, such that the training is done fast with far less queries, and subsequently, improves generalization. In essence, we achieve this by re-parameterizing prompts in low-rank representations in effective forms and designing intrinsic-dimensional clipping of zeroth-order gradients (Section 4).

Fundamentally, we are inspired by a line of previous works that hint at the dimension dependency of zeorth-order methods (Spall, 1992; Ghadimi & Lan, 2013; Duchi et al., 2015) and noise in stochastic methods (Bottou et al., 2018) causing optimization difficulty when it comes to large models. Indeed, we find this pertinent to BBPT in our empirical analysis, in which we show that a naive zeroth-order method suffers from increased dimensionality unlike the first-order counterpart (Section 3).

Our extensive experimental results show that ZIP is extremely efficient and robust, setting a new state of the art. To be specific, we evaluate ZIP on 13+ datasets for various vision-language tasks in standard benchmarks including few-shot learning, base-to-new generalization, cross-dataset transfer, and out-of-distribution generalization tasks, and across all, we find that ZIP consistently outperforms existing BBPT methods by substantial margins in terms of prediction accuracy, while demanding far less number of queries (Section 5). We provide a summary of how ZIP performs in Figure 1.

## 2   BACKGROUND

Prompt-tuning is an emerging paradigm to update large pre-trained models before utilizing them for various downstream tasks (Lester et al., 2021; Liu et al., 2023b). It works by prepending learnable context tokens to the input prompts embedding and training them on some data for a target task. Specifically, we can formulate it as an optimization problem as follows:

$$\min_{\theta} f(\theta, \omega; \mathcal{D}) \qquad (1)$$

where $\theta \in \mathbb{R}^d$ refers to the learnable parameters where $d = p \times m$ denotes the problem dimensionality with $p$ and $m$ being the word embedding dimensions and number of context tokens, respectively, and $\omega$ refers to the pre-trained model; also, $f$ refers to the loss function, which is the cross-entropy for vision-language tasks in our case, and $\mathcal{D}$ is a given dataset.

However, prompt-tuning is not directly applicable to *black-box* models since back-propagation is not allowed, and thus, the optimization must be done derivative-free. To this end, many approaches have been developed to address this issue for namely derivative-free optimization (Larson et al., 2019). In particular, a series of evolution strategies such as covariance matrix adaptation (Hansen, 2016) has been used for black-box prompt tuning or BBPT (Sun et al., 2022b;a; Yu et al., 2023).

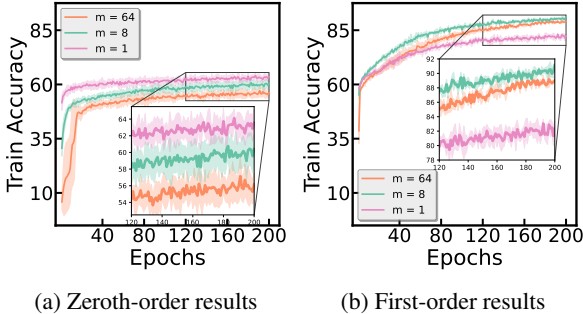

(a) Zeroth-order results     (b) First-order results

Figure 2: Zeroth-order vs. first-order methods for prompt tuning under varying number of prompt parameters. The training progresses are plotted measured on the Flowers102 dataset; we provide results on 10 other datasets in Figure 9 of Appendix B.1 where we observe the same trend that the zeroth-order method suffers from increasing prompt dimensionality.

An alternative approach to prompt-tuning in a black-box situation is by *zeroth-order* optimization which leverages approximate gradients estimated from only function evaluations. In particular, one of the foundational techniques is the simultaneous perturbation stochastic approximation (SPSA) (Spall, 1992; 1997), by which the gradient estimate is computed as follows:

$$\widehat{\nabla} f(\theta; \mathcal{B}) = \frac{1}{N} \sum_{i=1}^{N} \frac{f(\theta + cz_i; \mathcal{B}) - f(\theta - cz_i; \mathcal{B})}{2c} (z_i)^{-1}, \tag{2}$$

where $z_i$ refers to the perturbation vector randomly drawn from a probability distribution with zero mean and finite inverse moment, $(\cdot)^{-1}$ denotes element-wise reciprocal, and $N$ indicates the number of perturbation vector samples used for one gradient estimate; also, $c$ and $\mathcal{B}$ refer to a small positive scalar controlling the perturbation magnitude and the mini-batch of data points, respectively; *i.e.*, SPSA estimates gradients by simultaneously perturbing all dimensions in $\theta$. Then, the optimization is done iteratively using the zeroth-order stochastic gradient descent (ZO-SGD) (Ghadimi & Lan, 2013) with (2) as follows:

$$\theta_{t+1} = \theta_t - \eta_t \widehat{\nabla} f(\theta_t; \mathcal{B}_t), \tag{3}$$

where $\eta_t$ denotes the step size at iteration $t$.

This approach has been demonstrated to be effective for BBPT tuning of vision-language models (Oh et al., 2023; Tsai et al., 2020), and yet, in theory, zeroth-order methods can suffer from high variance and slow convergence, especially for high-dimensional problems (Spall, 1992; Ghadimi & Lan, 2013; Duchi et al., 2015). This means that one needs to query the model a high number of times, which we find is critical to secure the feasibility of BBPT in practice. As we show throughout this work, our key idea to fix this issue is to reduce the dimensionality of $\theta$ and the variance of $\widehat{\nabla} f$.

## 3  MOTIVATION

In this section, we motivate our work by disclosing that naively applying a zeroth-order method can lead to a failure of BBPT. Precisely, we show that its performance deteriorates as the number of context tokens (*i.e.*, dimension of trainable parameters) increases. This is rather unexpected because increasing their dimensions, in fact, is found to improve performance when optimized with a first-order method for prompt tuning.

To be specific, we optimize a vision-language model using both the basic zeroth-order or ZO (3) and first-order or FO (*i.e.*, SGD) methods, with varying the number of context tokens (1, 8, 64), and measure their training progress. We used CLIP (Radford et al., 2021), a representative pre-trained vision-language model widely used in the literature. The results are plotted in Figure 2. We summarize two key observations as follows:

- General prompt tuning can benefit from increased parameters, achieving higher expressive power and performance, especially with a moderate number of context tokens (*e.g.*, 8 tokens).

- In contrast, prompt tuning with zeroth-order optimization suffers in both training speed and performance as the number of context tokens increases.

Figure 3: Overview of ZIP framework.

These findings reveal a fundamental limitation of directly employing a zeroth-order method in BBPT: they are potentially applicable to BBPT, but not to the degree to which their performance is compatible with those of first-order methods or they can benefit from an increased number of context tokens.

Additionally, we further provide a convergence analysis of ZO-SGD (3) in Theorem 1 of Appendix A.2 to clearly show its dependency on the problem dimensionality $d$ and highlight the limitation of the zeroth-order approach.

We aim to alleviate this discrepancy in this work. This approach represents a significant departure from previous studies on BBPT, which have primarily focused on applying derivative-free optimization to specific domains, such as LLMs (Sun et al., 2022b) or vision-language models (Yu et al., 2023; Oh et al., 2023). We directly tackle this limitation of employing zeroth-order method for BBPT by enhancing both its efficiency and effectiveness, even suggesting a potential to bridge the gap between black-box and general prompt tunings.

## 4 ZIP: ZEROTH-ORDER INTRINSIC-DIMENSIONAL PROMPT-TUNING

In this section, we explain ZIP in detail, which is designed to address the fundamental issue of applying a zeroth-order method to BBPT. To this end, we first suggest reducing the number of learnable parameters in representing the context tokens via a series of reparameterization techniques. Then, we introduce a gradient clipping technique tailored for zeroth-order optimization in intrinsic dimensions to enhance the efficiency and effectiveness of ZIP. An overview of the ZIP framework is provided in Figure 3.

### 4.1 PROMPT TUNING IN LOWER DIMENSIONAL SPACES

As discussed in previous sections, the convergence, and hence the performance, of zeroth-order methods depends highly on the problem dimensionality (Ghadimi & Lan, 2013; Duchi et al., 2015). We thus first propose reducing the optimization dimensionality to secure applicability of zeorth-order methods to BBPT. In fact, we are also inspired by the concept of intrinsic dimension (Li et al., 2018; Aghajanyan et al., 2021) which refers to an effective dimensionality of a given problem and suggests that optimizing in that dimension can be as effective as in the full parameter space.

Specifically, we first project the learnable parameters of each context token $\theta_i \in \mathbb{R}^p$ onto a lower-dimensional space $\boldsymbol{v}_i \in \mathbb{R}^q$ using a Fastfood transform matrix $\mathbf{M}_i \in \mathbb{R}^{p \times q}$ as in Le et al. (2013); Li et al. (2018); Aghajanyan et al. (2021). The total number of trainable parameters are then reduced from $d = p \times m$, to $d' = q \times m$ with $d' \ll d$. Each vector indexed by $i$ corresponds to the $i$-th trainable context token where $m$ indicates the total number of context tokens. The random projection can then be expressed as follows:

$$\theta_i = \theta_{0,i} + \mathbf{M}_i \boldsymbol{v}_i \tag{4}$$

where $\theta_{0,i}$ is initial parameters. With this reparameterization, we can project learnable parameter to much lower dimension, from $d$ to $d'$.

Further, we apply another reparameterization in a low-rank approximation fashion as below, to get trainable parameter matrix $\mathbf{W}$:

$$\mathbf{W} = [\boldsymbol{v}_1'|\boldsymbol{v}_2'|\cdots|\boldsymbol{v}_m'] = \mathbf{U}\mathrm{diag}(\boldsymbol{s})\mathbf{V}^T. \tag{5}$$

We initialize $\mathbf{U}$ to zeros, $\mathbf{V}$ with standard normal distribution, and $s$ to ones, ensuring $\mathbf{W}$ starts at zero. With this reparameterization, the optimization variables become $\mathbf{U} \in \mathbb{R}^{q \times r}$, $\mathbf{V} \in \mathbb{R}^{r \times m}$, and $s \in \mathbb{R}^r$. Unlike the other conventional low-rank approximation methods (Hu et al., 2022), we interpose a diagonal matrix, which can effectively adjust the importance of specific dimensions by letting diagonal parameters directly scaling each dimension independently. We verify the effectiveness of our approach in experiments section 6.3 and also compare with other decomposition methods in Appendix B.5. This reparameterization allows us to reduce the number of trainable parameters while preserving the effectiveness as in optimizing the original parameters.

## 4.2 ENHANCING EXPRESSIVENESS WITH FEATURE SHARING

While reducing parameters can accelerate training with zeroth-order methods, it could also reduce the model expressivity. To address this issue, we introduce a feature sharing technique, which incorporates a vector $u \in \mathbb{R}^q$ within $\mathbf{W}$ and can serve as a common base across the partitioned vectors. This vector $u$ is integrated into original vectors $v'_i$, by using a outer product with a vector $\mathbb{1} \in \mathbb{R}^m$, forming the final trainable parameter matrix $\Xi$:

$$
\begin{aligned}
\Xi &= \mathbf{W} + u \otimes \mathbb{1} \\
&= \mathbf{U}\mathrm{diag}(s)\mathbf{V}^T + u \otimes \mathbb{1} \\
&= [v'_1 + u | v'_2 + u | \cdots | v'_m + u] \\
&= [w_1 | w_2 | \cdots | w_m]
\end{aligned}
\tag{6}
$$

where each $w_i \in \mathbb{R}^q$ is a mixed vector that blends the original components $v_i$ with the feature sharing by $u$. The updated parameters for context tokens are then computed as:

$$
\theta_i = \theta_{0,i} + \mathbf{M}_i w_i
\tag{7}
$$

We argue that the model can better learn complex features, leading to improved performance. This technique only requires a negligible amount of learnable parameters. We empirically validate the importance of the feature sharing in Section 6.2.

## 4.3 REDUCING VARIANCE WITH INTRINSIC-DIMENSIONAL CLIPPING

Through a series of reparameterization schemes, we obtained the final trainable parameters matrix $\Xi$, in which there are $\delta = r(q + m + 1) + q$ parameters in total. Now, the problem (1) reduces to

$$
\min_{\Xi} f(\Xi, \omega; \mathcal{D}).
\tag{8}
$$

One can consider employing ZO-SGD (3) to solve this problem, and yet, as demonstrated in Section 3, it can still cause slow convergence in practice due to its large variance.

To address this issue, we propose a simple yet robust zeroth-order method based on what we call intrinsic-dimensional gradient clipping mechanism defined as follows

$$
\Xi_{t+1} = \Xi_t - \eta_t \alpha_t \widehat{\nabla} f(\Xi_t, \omega; \mathcal{B}),
\tag{9}
$$

where $\alpha_t$ is a scaling factor defined as follows

$$
\alpha_t = \min\left( \frac{\sqrt{\delta}}{\sqrt{\sum_{i=1}^{\delta} \widehat{\nabla} f(\Xi_t, \omega; \mathcal{B})_i^2}}, 1 \right),
\tag{10}
$$

where $\delta$ refers to the problem dimensionality as mentioned above; *i.e.*, it clips the zeroth-order stochastic gradient estimates $\widehat{\nabla} f$ if its norm exceeds $\sqrt{\delta}$ as a threshold, while iteratively updating $\Xi_t$. There are several interesting aspects of this method as described below.

First, the immediate advantage of this approach is that there is no need to manually select the clipping threshold (which is prone to be suboptimal) or perform an expensive hyperparameter search. Considering that gradient clipping can accelerate the optimization process in general (Zhang et al., 2020b), and yet, that an appropriate choice of the threshold value is required, this advantage is certainly nontrivial. We validate this adaptivity by showing that the threshold chosen based on (10) is, quite surprisingly, nearly optimal across diverse training workloads in Section 6.1.

Also, the threshold being expressed in terms of the problem dimensionality $\delta$, in particular, $\sqrt{\delta}$, should be reasonably inspiring, considering research results in the literature. Specifically, we can interpret that previous work suggests setting the clipping threshold (for general first-order methods) to be the standard deviation of estimated gradients (Zhang et al., 2020a; Pascanu et al., 2012; Zhang et al., 2020b;c). While this is again not quite practical to compute, we notice that it can be done relatively straightforwardly for zeroth-order optimization, since the variance of zeroth-order gradients is inherently bounded in terms of the problem dimensionality $\delta$, thus the standard deviation being $\sqrt{\delta}$. We explicitly show this in Lemma 2 of Appendix A.1.

## 5 EVALUATIONS

### 5.1 EXPERIMENTAL SETUP

**Datasets and tasks.** To assess the query efficiency and performance of ZIP, we conduct evaluations on standard generalization tasks following the protocols of Zhou et al. (2022a;b); Oh et al. (2023). These tasks include few-shot learning, base-to-new generalization, cross-dataset transfer, and out-of-distribution (OOD) generalization. For few-shot learning, base-to-new generalization, and cross-dataset transfer, we evaluate ZIP across 13 diverse image classification tasks: ImageNet (Deng et al., 2009), Caltech101 (Fei-Fei et al., 2004), OxfordPets (Parkhi et al., 2012), Flowers102 (Nilsback & Zisserman, 2008), Food101 (Bossard et al., 2014), FGVCAircraft (Maji et al., 2013), SUN397 (Xiao et al., 2010), Resisc45 (Cheng et al., 2017), DTD (Cimpoi et al., 2014), SVHN (Netzer et al., 2011), EuroSAT (Helber et al., 2019), CLEVR (Johnson et al., 2017), and UCF101 (Soomro et al., 2012). For evaluating OOD generalization, we employ four established OOD datasets to measure the robustness of ZIP under distribution shifts: ImageNetV2 (Recht et al., 2019), ImageNet-Sketch (Wang et al., 2019), ImageNet-A (Hendrycks et al., 2021b), and ImageNet-R (Hendrycks et al., 2021a).

**Baselines.** To thoroughly evaluate the performance of ZIP, we compare it against a variety of baselines: (1) manual prompt, where manually composed prompts to conduct the evaluation (human-written prompts are detailed in Table 15); (2) state-of-the-art BBPT approaches for VLMs including BAR (Tsai et al., 2020), BLACKVIP (Oh et al., 2023) and BPTVLM (Yu et al., 2023). For all baselines, we follow the standardized few-shot evaluation protocol across datasets, consistent with Zhou et al. (2022b); Oh et al. (2023), which includes specific few-shot splits to ensure a fair comparison.

**Implementation details.** We mainly experiment using the CLIP (Radford et al., 2021) model with vision transformer (Dosovitskiy et al., 2021), keeping the CLIP model frozen. We consistently set the number of context tokens $m$ as 8 for ZIP and use 5,000 queries across all tasks for all BBPT baselines. The number of the intrinsic dimensionality $d'$ is set to 500, and the rank of low-rank matrices $r = 5$, resulting in a total of 417 learnable parameters $\delta$ with the formula $r(\lfloor d'/m \rfloor + m + 1) + \lfloor d'/m \rfloor$. Following the previous works for transfer learning (Zhou et al., 2022a;b; Oh et al., 2023), we initialize soft prompts from prompts derived from source tasks. We use the official code to reproduce BBPT baselines, and the results are averaged over three different random seeds. The implementation is available at https://github.com/LOG-postech/ZIP.

### 5.2 GENERALIZATION PERFORMANCE

We present empirical evidence showcasing the effectiveness and robustness of our proposed method, ZIP, across 13+ vision-language tasks. Our results, summarized in Table 1, 2, and 3, cover evaluations on few-shot accuracy, base-to-new generalization, and cross-dataset transfer with out-of-distribution generalization. The experiments reveal two main insights: (i) ZIP consistently outperforms other BBPT baselines across various tasks; (ii) ZIP achieves better robustness to unseen data distribution compared to existing BBPT methods; Detailed analyses of these findings are provided below.

**Few-shot performance.** As represented in Table 1, our experimental result indicates that ZIP consistently outperforms state-of-the-art BBPT approaches, including BAR, BLACKVIP, and BPTVLM, across 11 out of 13 datasets. On average, ZIP achieves accuracy gains of +7.7%, +6.8%, and +7.8% over BAR, BLACKVIP, and BPTVLM, respectively, demonstrating notable effectiveness in few-shot learning. In particular, ZIP excels on datasets requiring coarse semantic understanding, with

Table 1: Few-shot performance on 13 vision-language tasks. All the results are based on 16-shots per class. The **bold numbers** denote the highest accuracy of all baselines on each dataset, and the underlined values indicate the second. ZIP clearly outperforms other BBPT baselines.

| Method | #Params | Caltech101 | OxfordPets | Flowers102 | Food101 | FGVCAircraft | SUN397 | DTD | SVHN | EuroSAT | Resisc45 | CLEVR | UCF101 | ImageNet | Average |
|---|---|---|---|---|---|---|---|---|---|---|---|---|---|---|---|
| Manual Prompt | 0k | 93.2 | 89.1 | 70.8 | 85.9 | 24.8 | 62.6 | 44.1 | 19.2 | 48.4 | 57.2 | 15.2 | 67.5 | 66.7 | 57.3 |
| BAR | 37.6k | 92.5 | 85.6 | 65.0 | 83.0 | 21.6 | 62.4 | 42.9 | 19.8 | 51.6 | 53.9 | 18.1 | 63.5 | 64.0 | 55.7 |
| BLACKVIP | 9.9k | 92.6 | 86.9 | 63.5 | 83.5 | 21.5 | 62.3 | 43.1 | 27.5 | 44.4 | 55.5 | 25.9 | 64.0 | 65.5 | 56.6 |
| BPTVLM | 4.0k | 88.6 | 89.4 | 66.9 | 84.2 | 24.0 | 53.2 | 40.6 | 29.8 | 53.0 | 56.2 | 16.4 | 64.8 | 55.5 | 55.6 |
| ZIP | 0.4k | 94.0 | 92.3 | 70.4 | 86.4 | 26.8 | 63.3 | 47.8 | 47.8 | 64.6 | 66.1 | 28.4 | 69.8 | 66.2 | 63.4 |

Table 2: Base-to-new generalization performance. H represents the harmonic mean, providing a balanced measure of accuracy across seen and unseen classes (Xian et al., 2017). ZIP consistently outperforms BAR, BLACKVIP, and BPTVLM across base, new, and harmonic mean evaluations.

| Method | Set | Caltech101 | OxfordPets | Flowers102 | Food101 | FGVCAircraft | SUN397 | DTD | SVHN | EuroSAT | Resisc45 | CLEVR | UCF101 | ImageNet | Average |
|---|---|---|---|---|---|---|---|---|---|---|---|---|---|---|---|
| BAR | Base | 96.5 | 87.3 | 67.5 | 87.5 | 25.6 | 69.2 | 51.2 | 23.5 | 60.2 | 66.1 | 27.5 | 66.4 | 69.6 | 61.4 |
| BLACKVIP | Base | 96.6 | 87.7 | 67.9 | 87.6 | 25.8 | 69.0 | 51.8 | 26.4 | 66.4 | 69.9 | 38.9 | 67.0 | 70.3 | 63.5 |
| BPTVLM | Base | 93.2 | 90.6 | 66.9 | 88.7 | 29.1 | 65.3 | 53.2 | 45.4 | 70.3 | 72.0 | 41.5 | 68.3 | 66.3 | 65.4 |
| ZIP | Base | 97.0 | 94.9 | 72.1 | 89.9 | 29.7 | 70.3 | 61.7 | 52.9 | 84.0 | 81.6 | 50.1 | 75.1 | 72.1 | 71.6 |
| BAR | New | 94.5 | 94.9 | 73.2 | 88.9 | 29.2 | 74.6 | 55.8 | 27.3 | 72.1 | 62.3 | 27.1 | 73.3 | 64.9 | 64.5 |
| BLACKVIP | New | 93.2 | 90.9 | 74.5 | 89.4 | 30.9 | 73.9 | 55.4 | 21.8 | 48.8 | 61.2 | 28.0 | 72.6 | 66.8 | 62.1 |
| BPTVLM | New | 92.7 | 95.8 | 72.7 | 85.4 | 32.3 | 64.8 | 45.3 | 40.1 | 47.0 | 61.3 | 28.4 | 65.0 | 55.2 | 60.5 |
| ZIP | New | 94.3 | 97.0 | 73.4 | 90.0 | 32.9 | 71.5 | 51.0 | 45.8 | 64.4 | 65.2 | 26.8 | 69.5 | 65.6 | 65.2 |
| BAR | Harmonic | 95.5 | 90.9 | 70.2 | 88.2 | 27.3 | 71.8 | 53.4 | 25.3 | 65.6 | 64.1 | 27.3 | 67.7 | 67.2 | 62.9 |
| BLACKVIP | Harmonic | 94.9 | 89.3 | 71.0 | 88.5 | 28.1 | 71.4 | 53.5 | 23.9 | 56.3 | 65.3 | 32.6 | 69.7 | 68.5 | 62.8 |
| BPTVLM | Harmonic | 92.9 | 93.1 | 69.7 | 87.0 | 30.6 | 65.0 | 48.9 | 42.6 | 56.3 | 66.2 | 33.7 | 66.6 | 60.2 | 62.9 |
| ZIP | Harmonic | 95.6 | 95.9 | 72.7 | 89.9 | 31.2 | 70.9 | 55.8 | 49.1 | 72.9 | 72.5 | 34.9 | 72.2 | 68.7 | 67.9 |

Table 3: Cross-dataset transfer and out-of-distribution generalization performance. After training on ImageNet (*i.e.*, source) with 16-shot data per class, ZIP is evaluated on 12 target datasets for CDT and 4 ImageNet variants for OOD. ZIP demonstrates better transferability and generalizability, outperforming BAR, BLACKVIP, and BPTVLM.

| | Source | CDT Target | | | | | | | | | | | | | OOD Target | | | | |
|---|---|---|---|---|---|---|---|---|---|---|---|---|---|---|---|---|---|---|---|
| Method | ImageNet | Caltech101 | OxfordPets | Flowers102 | Food101 | FGVCAircraft | SUN397 | DTD | SVHN | EuroSAT | Resisc45 | CLEVR | UCF101 | Average | ImageNet-A | ImageNetV2 | ImageNet-R | ImageNet-Sketch | Average |
| BAR | 64.0 | 92.3 | 84.3 | 64.3 | 83.1 | 20.8 | 61.0 | 42.2 | 20.0 | 49.6 | 50.6 | 14.5 | 63.0 | 53.8 | 40.2 | 57.5 | 72.0 | 43.8 | 53.4 |
| BLACKVIP | 65.5 | 92.5 | 86.2 | 64.9 | 83.6 | 22.3 | 62.0 | 43.3 | 18.7 | 55.7 | | 15.2 | 64.1 | 54.1 | 42.5 | 59.2 | 73.1 | 44.6 | 54.9 |
| BPTVLM | 55.5 | 80.7 | 77.7 | 50.3 | 77.6 | 16.3 | 43.8 | 30.8 | 15.5 | 34.6 | 37.7 | 12.4 | 54.8 | 44.4 | 32.7 | 46.7 | 61.7 | 33.5 | 43.7 |
| ZIP | 66.2 | 92.4 | 85.7 | 65.9 | 84.6 | 20.5 | 60.1 | 41.5 | 26.4 | 46.7 | 54.7 | 17.4 | 61.3 | 54.8 | 47.1 | 59.7 | 75.2 | 45.5 | 56.9 |

improvements of +11.6% on EuroSAT, and +8.9% on Resisc45 compared to the second-best method. Additionally, ZIP shows remarkable performance in digit recognition, surpassing the next best method by +18.0% on the SVHN dataset, further highlighting its capability in few-shot learning.

**Base-to-new generalization.** Table 2 presents the base-to-new generalization results, where models are trained on base classes and evaluated on both base and new classes across 13 datasets. ZIP consistently outperforms all BBPT baselines, achieving the highest base, new, and harmonic mean scores. By leveraging its lower parameter count and the robustness of zeroth-order optimization, ZIP effectively mitigates overfitting, as its reduced model capacity and rough gradient estimates help avoid fitting to noisy outliers, resulting in better generalization.

**Cross-dataset transfer & Out-of-distribution generalization.** To assess robustness in challenging scenarios, we evaluate ZIP for cross-dataset transfer (CDT) and out-of-distribution (OOD) generalization. As shown in Table 3, ZIP, trained on ImageNet (*i.e.*, source), demonstrates competitive generalization capabilities in the CDT setting, achieving slight improvements of 1.0% over BAR and

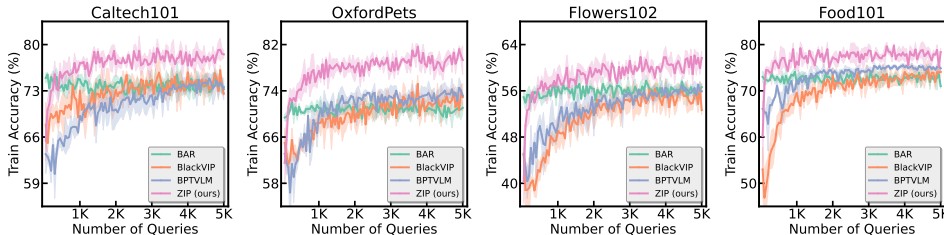

Figure 4: Training performance measured on Caltech101, OxfordPets, Flowers102, and Food101. We provide more results on other datasets in Figure 10.

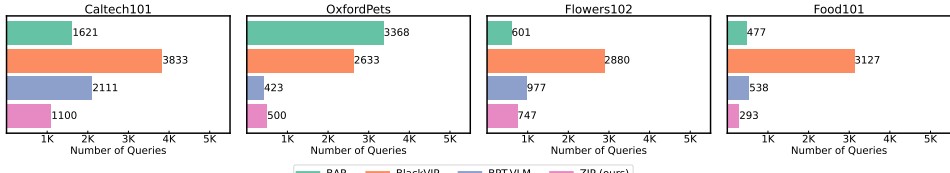

Figure 5: Number of queries to reach a target accuracy. For the datasets Caltech101, OxfordPets, Flowers102, Food101, and FGVCAircraft, ZIP requires fewer API calls to reach target accuracy thresholds in most cases. This demonstrates considerable query efficiency when compared to other BBPT methods. We provide more results on other datasets in Figure 11.

0.7% over BLACKVIP, with a more notable gain of 10.4% over BPTVLM across 12 diverse target datasets. More significantly, in the OOD evaluations on four ImageNet variants, ZIP consistently outperforms all baselines, achieving substantial gains of 3.5% over BAR, 2.0% over BLACKVIP, and a remarkable 13.2% improvement over BPTVLM. These results highlight the exceptional robustness and adaptability of ZIP in handling domain shifts, making it particularly effective for real-world applications where OOD generalization is critical.

## 5.3 QUERY EFFICIENCY

This section provides empirical evidence highlighting the query efficiency of ZIP. We start by tracking the training progress of different BBPT methods, ensuring all operate within the same computational budget. For a fair comparison, ZIP and other baselines are allocated a budget of 5,000 queries. This query budget was chosen to reflect a more practical scenario, as many existing methods often require thousands of epochs (Oh et al., 2023), which is unrealistic for real-world applications with strict API query limitations. By setting a more feasible budget, we aim to evaluate efficiency of each methods under conditions that closely resemble practical deployment settings.

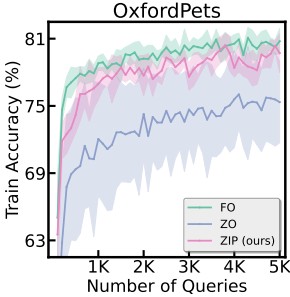

Figure 6: Training curves of first-order (FO), zeroth-order (ZO) and ZIP. ZIP effectively bridges the gap between FO and ZO with notably faster training and high accuracy.

Figure 4 shows that ZIP consistently achieves faster training speed and higher accuracy than other BBPT methods under identical query budget constraints. This efficiency is attributed to the effective combination of low-rank approximation with diagonal matrix and our specialized threshold for zeroth-order optimization, which accelerates training, while the feature sharing and compactness of the low-rank representation enhance overall performance. These design elements work synergistically, allowing ZIP to achieve rapid training progress and improved accuracy. Detailed analysis of the module combination is provided in Appendix B.7. To further analyze query efficiency, we evaluate the number of queries required to reach target accuracy, which is determined as the minimum of the maximum accuracy achieved by all methods. As shown in Figure 5, ZIP demonstrates strong query efficiency, achieving the best performance in datasets like Caltech101 and Food101, and maintaining competitive efficiency in OxfordPets and Flowers102, even when not the absolute best. The overall results, summarized in Figure 1a, show that ZIP achieves over a 48% improvement in query efficiency compared to the second-best BBPT method. This indicates that ZIP utilizes its query budget effectively, making it particularly suited in resource-constrained scenarios.

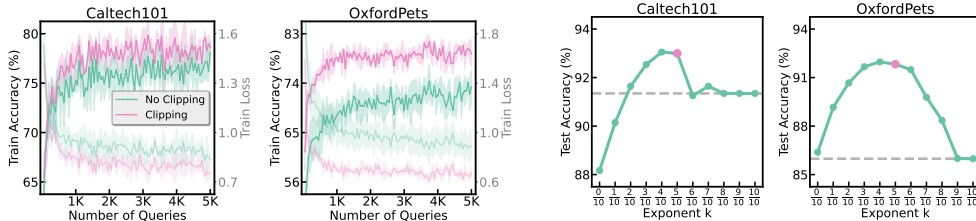

(a) Effects of our clipping threshold (in Section 6.1)    (b) clipping method varying threshold $\delta^k$

Figure 7: Effects of intrinsic-dimensional clipping. (a) Training progress comparison with and without intrinsic-dimensional clipping. (b) Test accuracy with varying thresholds $\delta^k$. The red point indicates the chosen threshold of ZIP, which consistently achieves near-optimal accuracy.

Table 4: Benefits of feature sharing over unshared. Integrating shared features consistently boosts model expressive power and accuracy across diverse tasks, demonstrating improved performance.

| Method | Caltech101 | OxfordPets | Flowers102 | Food101 | FGVCAircraft | SUN397 | DTD | SVHN | EuroSAT | Resisc45 | CLEVR | UCF101 | ImageNet | Average |
|---|---|---|---|---|---|---|---|---|---|---|---|---|---|---|
| Unshared | 93.1 | 90.8 | 67.1 | 86.0 | 25.2 | 59.0 | 44.4 | 40.9 | 60.6 | 63.3 | 20.2 | 67.4 | 65.2 | 60.2 |
| Shared | **93.5** | **91.8** | **70.6** | **86.2** | **26.3** | **62.2** | **46.5** | **43.8** | **66.2** | **65.6** | **24.4** | **69.0** | **65.5** | **62.4** |

We also compare the query efficiency of ZIP with first-order and naive zeroth-order optimization, using 8 tokens and 5,000 queries across all methods. As shown in Figure 6, ZIP bridges the gap between first-order and zeroth-order optimization, achieving training speeds similar to first-order on the OxfordPets dataset. While zeroth-order methods typically exhibit dependence on $d$ for training speed, the efficient design of ZIP allows it to match first-order optimization behavior. This demonstrates the enhanced query efficiency of ZIP, making it highly suitable for practical applications where efficient resource utilization is critical. Further details on query efficiency across additional datasets can be found in Figure 10, 11 and 12.

## 6 ABLATIONS

### 6.1 INTRINSIC-DIMENSIONAL CLIPPING

In this section, we evaluate the effectiveness of our clipping method, with setting threshold as $\sqrt{\delta}$. We begin by tracking the training progress of ZIP with intrinsic-dimensional clipping and the one without. As shown in Figure 7a, ZIP with our clipping threshold consistently achieves faster training speeds and higher accuracy, indicating its efficiency in enhancing zeroth-order optimization. This improvement is largely due to the variance-reducing nature of clipping, which results in more stable gradient estimates and consequently accelerates the training process.

To further validate the effectiveness of gradient clipping with our threshold, we compared $\sqrt{\delta}$ threshold against various alternative values to ensure its optimality. As shown in Figure 7b, the $\sqrt{\delta}$ threshold consistently achieved near-optimal performance on Caltech101 and OxfordPets, outperforming other clipping settings ranging from 1 ($= \delta^{0/10}$) to $\delta$ ($= \delta^{10/10}$). The gray dashed line, representing no clipping, further underscores the advantage of $\sqrt{\delta}$ threshold. These results highlight the effectiveness of the $\sqrt{\delta}$ threshold, demonstrating its capability as an efficient clipping strategy for zeroth-order optimization without requiring extensive hyperparameter tuning. Additional validation results on other datasets are available in Figure 16 and 17.

### 6.2 FEATURE SHARING

To evaluate the expressive power of feature sharing, we compared the performance of models with and without feature sharing. As shown in Table 4, models utilizing feature sharing consistently achieved higher accuracy, increasing the overall average score from 60.2% to 62.4%. These consistent gains

Table 5: Benefits of low-rank approximation with diagonal matrix. Comparing our method against standard dimensionality reduction, demonstrating notable test accuracy improvements.

| Method | Caltech101 | OxfordPets | Flowers102 | Food101 | FGVCAircraft | SUN397 | DTD | SVHN | EuroSAT | Resisc45 | CLEVR | UCF101 | ImageNet | Average |
|---|---|---|---|---|---|---|---|---|---|---|---|---|---|---|
| Standard | 90.9 | 88.1 | **67.5** | 84.6 | 23.8 | 57.9 | 43.2 | 31.5 | 56.5 | 58.3 | 18.3 | 65.3 | 62.3 | 57.6 |
| Ours | **93.1** | **90.8** | 67.1 | **86.0** | **25.2** | **59.0** | **44.4** | **40.9** | **60.6** | **63.3** | **20.2** | **67.4** | **64.8** | **60.2** |

across diverse datasets highlight the effectiveness of features sharing in retaining model expressiveness and improving performance, even when parameters are reduced.

Furthermore, feature sharing shows consistent benefits across diverse datasets, underscoring its robustness as a technique for maintaining accuracy while optimizing parameter efficiency. To evaluate whether feature sharing improves generalization to unseen datasets, we analyze its impact on base-to-new generalization, cross-dataset transfer, and out-of-distribution scenarios, as detailed in Appendix B.4. These findings validate the role of feature sharing in enhancing generalization and its potential utility in broader domain adaptation tasks.

### 6.3 Low rank approximation with diagonal matrix

The low-rank approximation with a diagonal matrix is pivotal in enhancing both the efficiency and performance of our method. Unlike naive lower-dimensional projections, this approach effectively preserves the most crucial components of the parameter space, allowing for accelerated training without compromising the expressive power of the model.

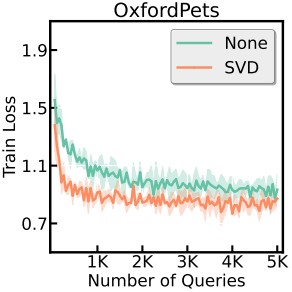

As shown in Figure 8 and Table 5, this approach not only accelerates the training process but also improves model accuracy. For instance, the average accuracy across datasets increased from 57.6% to 60.2% with the application of low-rank approximation using a diagonal matrix. These gains highlight the effectiveness of the technique in enhancing training efficiency and overall model performance, making it particularly advantageous for optimizing zeroth-order based prompt tuning compared to more straightforward projection methods. Additional results on other datasets further validating this improvement can be found in Figure 14.

Figure 8: Effects of low-rank approximation with diagonal matrix. Our method improves training efficiency compared to standard dimensionality reduction.

## 7 Conclusion

In this paper, we propose ZIP, a new method for prompt-tuning black-box vision-language models. Extensive experiments show that ZIP outperforms state-of-the-art BBPT methods in generalization performance while offering faster training with far less number of queries. We believe that our work unlocks numerous opportunities for future work including, for instance, extending to a broader range of foundation models and addressing diverse prompting schemes in different black-box optimization scenarios. We intend to explore these ideas in future work.

### Reproducibility statement

To ensure reproducibility, we provide detailed information on our experimental setup in Appendix C.4, including training and evaluation procedures. All datasets used in this work are publicly available. We conduct our experiments on NVIDIA 3090, A6000, A100 and Intel Gaudi-v2 GPUs.

ACKNOWLEDGEMENT

This work was partly supported by the Institute of Information & communications Technology Planning & Evaluation (IITP) grant funded by the Korean government (MSIT) (RS-2019-II191906, Artificial Intelligence Graduate School Program (POSTECH), RS-2024-00338140, Development of learning and utilization technology to reflect sustainability of generative language models and up-to-dateness over time, and RS-2022-II220959, (part2) Few-Shot learning of Causal Inference in Vision and Language for Decision Making), the National Research Foundation of Korea (NRF) grant funded by the Korea government (MSIT) (RS-2023-00210466, RS-2023-00265444), Samsung Research (IO240508-09825-01), and the NAVER-Intel Co-Lab. The work was conducted by POSTECH and reviewed by both NAVER and Intel.

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

# A  THEORETICAL ANALYSIS

## A.1  ASSUMPTION & LEMMA

**Assumption 1.** *On the function $f(\cdot)$, there exists some $L > 0$ such that for all $x, y$, we have $\|\nabla f(x) - \nabla f(y)\| \leq L\|x - y\|$*

**Lemma 1.** *(Unbiasdness of ZO-SGD) In the $c_t \to 0$ limit, ZO-SGD is a unbiased estiamtor of FO-SGD in terms of random perturbation vector, which follows a Bernoulli distribution of two different values with equal absolute value and probability. That is,*

$$\mathbb{E}_{\{z_n\}_{n=1}^N}(\widehat{\nabla} f(\theta_t; \mathcal{B}_t)) = \nabla f(\theta_t; \mathcal{B}_t) \tag{11}$$

*Proof of Lemma 1.* Note that as $c_t \to 0$ limit, we have

$$\widehat{\nabla} f(\theta_t; \mathcal{B}_t) = \frac{1}{N} \sum_{i=1}^N (z_i)^{-1} (z_i)^\top \nabla f(\theta_t; \mathcal{B}_t)$$

let $A^k \in R^{d \times d}$ be a matrix of $(z_k)^{-1}(z_k)^\top$, then we get

$$A_{ij}^k = \begin{cases} 1 & \text{if } i = j \\ \frac{z_{kj}}{z_{ki}} & \text{otherwise} \end{cases}$$

Note that $z_{ni}$ is a $i$-th element for the vector $z_n$. By taking expectation in terms of $z_n$ over matrix $A$, we can get

$$\mathbb{E}_{\{z_n\}_{n=1}^N}(A_{ij}^n) = \begin{cases} 1 & \text{if } i = j \\ 0 & \text{otherwise} \end{cases}$$

since $z_t^n$ have zero inverse moment and zero mean as we assumed. Therefore,

$$\mathbb{E}_{\{z_n\}_{n=1}^N}(\widehat{\nabla} f(\theta_t; \mathcal{B}_t)) = \nabla f(\theta_t; \mathcal{B}_t)$$

as desired. $\qquad\square$

**Lemma 2.** *(Second moment of ZO-SGD) In the $c_t \to 0$ limit, second moment of ZO-SGD in terms of random perturbation vector, which follows a Bernoulli distribution of two different values with equal absolute value and probability. That is,*

$$\mathbb{E}_{\{z_n\}_{n=1}^N}(\|\widehat{\nabla} f(\theta_t; \mathcal{B}_t)\|^2) = \frac{d}{N}\|\nabla f(\theta_t; \mathcal{B}_t)\|^2 \tag{12}$$

*Proof of Lemma 2.* Starting from Lemma 1, zeroth-order gradient can be represented as below.

$$\widehat{\nabla} f(\theta_t; \mathcal{B}_t) = \frac{1}{N} \sum_{n=1}^N A^n \nabla f(\theta_t; \mathcal{B}_t)$$

Therefore, the second moment of zeroth-order gradient

$$\mathbb{E}_{\{z_n\}_{n=1}^N}(\|\widehat{\nabla} f(\theta_t; \mathcal{B}_t)\|^2) = \mathbb{E}_{\{z_n\}_{n=1}^N}\left(\frac{1}{N} \sum_{n=1}^N \nabla f(\theta_t; \mathcal{B}_t)^\top (A^n)^\top A^n \nabla f(\theta_t; \mathcal{B}_t)\right)$$

let $B^n \in R^{d \times d}$ be a result of $(A^n)^\top A^n$, we can get

$$B_{ij}^n = \begin{cases} d & \text{if } i = j \\ \sum_{k=1}^d \frac{(z_{ni})^2}{z_{nj} z_{ni}} & \text{otherwise} \end{cases}$$

Taking expectation over matrix $B^n$, we can get

$$\mathbb{E}_{\{z_n\}_{n=1}^N}(B_{ij}^n) = \begin{cases} d & \text{if } i = j \\ 0 & \text{otherwise} \end{cases}$$

By plugging above results, the second moment of zeroth-order gradient is

$$\mathbb{E}_{\{z_n\}_{n=1}^N}(\|\widehat{\nabla} f(\theta_t; \mathcal{B}_t)\|^2) = \frac{d}{N}\|\nabla f(\theta_t; \mathcal{B}_t)\|^2.$$

as desired.

$\qquad\square$

**Lemma 3.** *With assumption 1, for any unbiased gradient estimate $\widehat{\nabla}f(\theta_t; z, \mathcal{B}_t)$,*

$$\mathbb{E}(f(\theta_{t+1})|x_t) \leq f(\theta_t) - \eta\|\nabla f(\theta_t)\|^2 + \frac{L}{2}\eta^2 \left\|\widehat{\nabla}f(\theta_t; z, \mathcal{B}_t)\right\|^2$$

### A.2 CONVERGENCE ANALYSIS OF ZEROTH-ORDER OPTIMIZATION

The high variance of zeroth-order gradient estimates stems from the estimation process involving random perturbations, introducing an additional problem dimension ($d$) related terms in convergence compared with corresponding first-order (FO) methods. Although the convergence rate was originally proven by Ghadimi & Lan (2013), we have also confirmed similar convergence behavior using Spall (1992) approach. Note that we assumed $z_i$ has zero inverse moment, as it is sampled from a Bernoulli distribution of two different values with equal absolute value and probability in practice. The convergence rate of ZO-SGD using (2) is as follows:

**Theorem 1** (Convergence rate of ZO-SGD). *Under Assumption 1, in the $c_t \to 0$ limit, when $\eta = \sqrt{\frac{2NF}{LGd}}\sqrt{\frac{1}{T}}$ where $F := f(x_0) - f(x_*)$ and sampling the $z_n$ from a Bernoulli distribution of two different values with equal absolute value and probability convergence rate of ZO-SGD is*

$$\frac{1}{T}\sum_{t=0}^{T-1}\mathbb{E}_{t,\{z_n\}_{n=1}^N}\|\widehat{\nabla}f(\theta_t; \mathcal{B}_t)\|_2^2 = \mathcal{O}\left(\sqrt{\frac{d}{T}}\right). \tag{13}$$

*Proof of Theorem 1.* With Lemma 1, we can start from Lemma 3. By assuming that FO-SGD has finite variance bound as $\mathbb{E}_t\left[\left\|\widetilde{\nabla}f(x_t)\right\|_2^2\right] \leq G$ and reformulate Lemma 3 then we get :

$$\|\nabla f(x_t)\|_2^2 \leq \frac{1}{\eta}\mathbb{E}_{t,\{z_n\}}\left[f(x_t) - f(x_{t+1})\right] + \frac{Ld}{2N}\eta G.$$

Summing over from $t = 0$ to $t = T$ :

$$\sum_{t=0}^{T-1}\|\nabla f(x_t)\|_2^2 \leq \frac{1}{\eta}\left[f(x_0) - \mathbb{E}f(x_T)\right] + \frac{Ld}{2N}\eta GT.$$

Remind that $f$ is lower bounded with $f_*$ and divide with $T$ :

$$\frac{1}{T}\sum_{t=0}^{T-1}\|\nabla f(x_t)\|_2^2 \leq \frac{f(x_0) - f_*}{\eta T} + \frac{Ld}{2N}\eta G.$$

Let $\eta = \mathcal{O}\left(\sqrt{\frac{1}{dT}}\right)$ then,

$$\frac{1}{T}\sum_{t=0}^{T-1}\|\nabla f(x_t)\|_2^2 = \mathcal{O}\left(\sqrt{\frac{d}{T}}\right).$$

$\square$

# B  FURTHER ANALYSIS

We conducted additional supplementary experiments to further validate and gain deeper insights into our proposed method, ZIP. To ensure a comprehensive analysis, we extended our evaluation to include all remaining standard classification tasks mentioned in Section 5 and 6. This extended evaluation provides a more detailed understanding of the performance of ZIP across a diverse range of datasets.

## B.1  IMPACT OF OPTIMIZATION METHODS VARYING CONTEXT TOKEN COUNTS

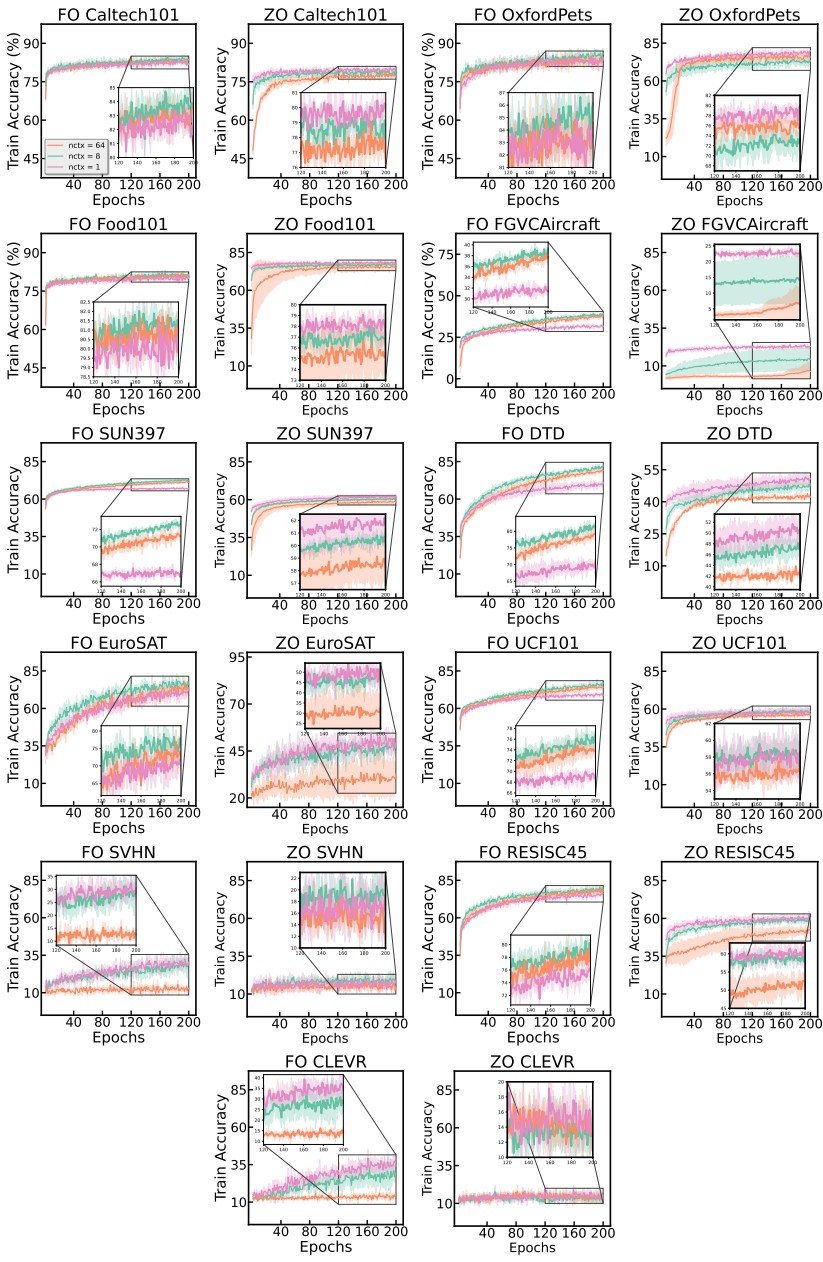

Figure 9: Effect of optimization methods in prompt tuning across various vision-language tasks.

We conduct a series of experiments to examine how varying the number of context tokens affects both first-order and zeroth-order optimization methods across multiple datasets, as illustrated in Figure 2 and 9.

Our findings reveal that zeroth-order optimization generally performs better with fewer context tokens (*e.g.*, 1 token). However, certain datasets such as UCF101, SVHN, and CLEVR deviate from this trend. In contrast, first-order optimization typically aligns with the trends shown in Section 3, displaying improved accuracy with a moderate number of context tokens across most datasets, except for SVHN and CLEVR, which demonstrate variations in optimal token counts.

These results suggest that while the optimal number of context tokens depends on the dataset, first-order optimization generally benefits from a larger context token count, whereas zeroth-order optimization tends to be more effective with fewer tokens.

## B.2 QUERY EFFICIENCY

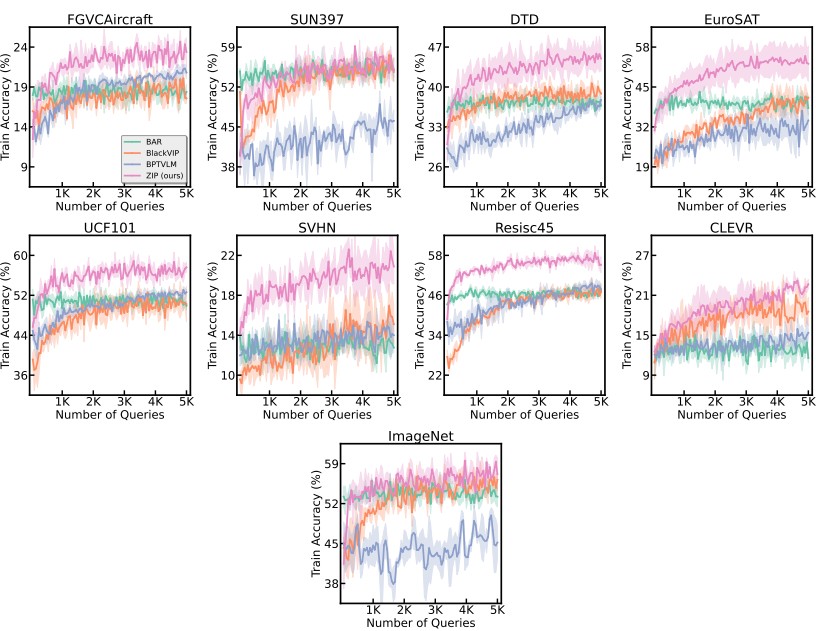

Figure 10: Training curves with 5,000 query budgets across various vision-language tasks.

Figure 4 and 10 display the training accuracy curves of ZIP under a 5,000 query budget across various tasks. Throughout the training process, ZIP consistently demonstrates faster training speeds and achieves higher accuracy compared to other BBPT methods across most datasets, highlighting its capability to utilize the available query budget more efficiently.

In Figure 5 and 11, we further analyze the number of API calls required to reach specific accuracy targets across various datasets. The target accuracy is determined as the minimum of the maximum accuracy achieved by all methods. The results indicate that ZIP consistently reaches these accuracy milestones with fewer queries than other methods, underscoring its query-efficient design and adaptability across a diverse range of tasks.

Additionally, in Figure 6 and 12, we compare the performance of first-order, zeroth-order optimization, and ZIP across multiple datasets. These results further validate our claim in Section 5.3 that ZIP effectively bridges the gap between first-order and zeroth-order optimization. ZIP not only consistently outperforms standard zeroth-order methods in test accuracy across all evaluated datasets but also frequently surpasses first-order optimization, demonstrating its outstanding training efficiency.

Moreover, we include results for context token $m = 1$ as a reference (See Figure 13), demonstrating that naive zeroth-order optimization with one token often struggles to match the performance of ZIP with 8 tokens, particularly in maintaining stable training accuracy. ZIP significantly outperforms the naive method on OxfordPets, FGVCAircraft, EuroSAT, and CLEVR. While the naive method shows comparable results on some other datasets, it is worth noting that even the first-order method with 8 tokens does not yield substantial improvements over the first-order method with 1 token on Caltech101, OxfordPets, and Food101 (See Figure 9). Additionally, using 1 token performs better on

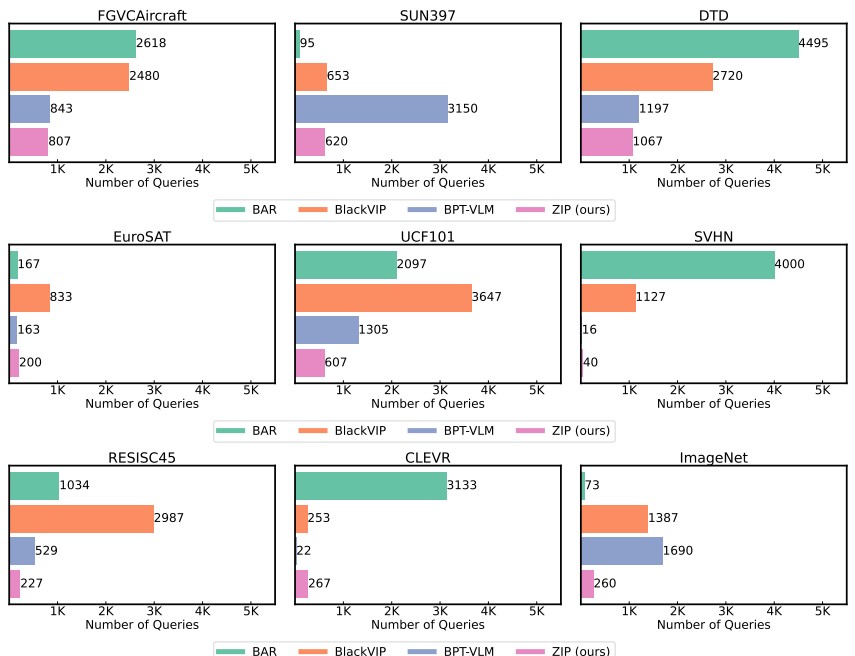

Figure 11: Queries to reach target accuracy across various vision-language tasks.

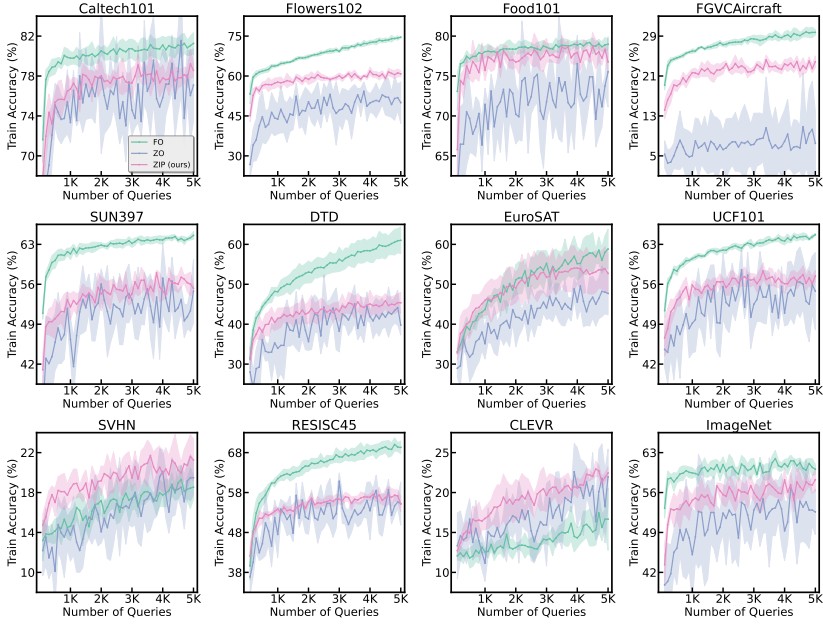

Figure 12: Training curves of first-order, zeroth-order and ZIP across various vision-language tasks.

CLEVR and SVHN, highlighting that the optimal number of prompt tokens remains an important factor for performance.

We also conduct an extended evaluation of the performance of ZIP by increasing the API query budget to 20,000, as detailed in Table 6. Several notable insights emerged from this analysis. With a 5,000-query budget, ZIP achieves an ImageNet accuracy of 66.2%, performing on par with strong baselines such as Manual Prompt (66.7%) and BLACKVIP (65.5%). When the query budget is increased to 20,000, ZIP further improves its accuracy to 67.2%, surpassing Manual Prompt and BLACKVIP by margins of 0.5% and 0.9%, respectively. These results highlight the ability of ZIP

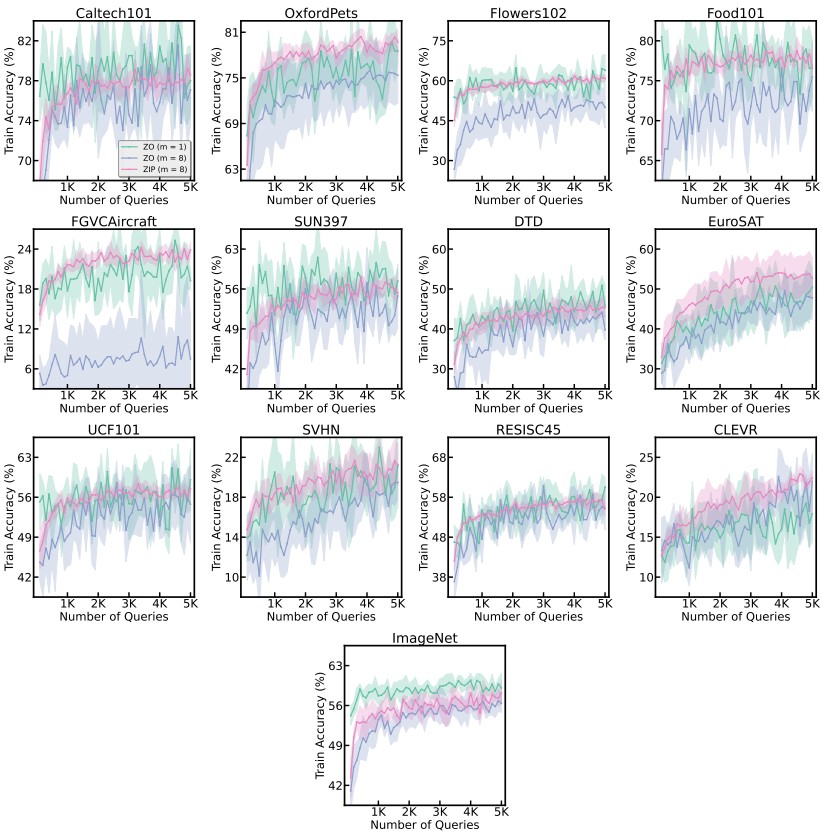

Figure 13: Training curves of zeroth-order ($m = 1$), zeroth-order ($m = 8$) and ZIP ($m = 8$) across various vision-language tasks.

Table 6: Few-shot performance on 13 vision-language tasks with 20,000 API query budget. All the results are based on 16-shots per class. The **bold numbers** denote the highest accuracy of all baselines on each dataset, and the underlined values indicate the second. ZIP consistently outperforms other BBPT baselines, achieving outstanding accuracy across diverse tasks and showcasing its scalability and efficiency under an expanded query budget.

| Method | #Params | Caltech101 | OxfordPets | Flowers102 | Food101 | FGVCAircraft | SUN397 | DTD | SVHN | EuroSAT | Resisc45 | CLEVR | UCF101 | ImageNet | Average |
|---|---|---|---|---|---|---|---|---|---|---|---|---|---|---|---|
| Manual Prompt | 0k | 93.0 | 89.1 | 70.6 | 85.9 | 24.8 | 62.6 | 44.1 | 18.8 | 48.1 | 58.1 | 14.5 | 67.5 | 66.7 | 57.2 |
| BAR | 37.6k | 92.5 | 88.1 | 67.7 | 83.2 | 23.0 | 63.8 | 47.1 | 32.3 | 54.0 | 64.1 | 25.4 | 65.7 | 65.5 | 59.4 |
| BLACKVIP | 9.9k | 93.0 | 88.0 | 64.8 | 85.0 | 22.5 | 62.3 | 44.4 | 42.3 | 57.2 | 57.1 | **28.8** | 66.1 | 66.3 | 59.8 |
| BPTVLM | 4.0k | 91.6 | 90.3 | 69.9 | 85.1 | 26.3 | 57.2 | 50.1 | 34.4 | 65.8 | 63.6 | 27.8 | 67.4 | 61.6 | 60.9 |
| **ZIP** | 0.4k | **94.1** | **92.4** | **71.8** | **86.9** | **27.3** | **64.4** | **52.9** | 49.9 | **66.6** | **68.4** | 28.6 | **70.0** | **67.2** | **64.7** |

to leverage additional query budgets effectively, scaling performance significantly with increased resources. Notably, compared to previous work, such as BLACKVIP, which achieved 67.1% on ImageNet using substantially higher query budgets (625,000 API queries), ZIP delivers competitive and often outstanding performance using only 20,000 API queries. This demonstrates exceptional efficiency and adaptability of ZIP, particularly in computationally constrained environments.

These supplementary findings reinforce our assertions in Section 5.3, confirming that ZIP not only accelerates training but also makes highly efficient use of query budgets, making it exceptionally suited for resource-constrained scenarios.

Table 7: Few-shot performance on 13 vision-language tasks with SigLIP (Zhai et al., 2023). All results are based on 16-shot data per class. **Bold numbers** represent the highest accuracy among all baselines for each dataset, while underlined values indicate the second-best. On average, ZIP outperforms other BBPT baselines, demonstrating its strong generalization and adaptability across diverse datasets, even when applied to a vision-language model distinct from CLIP.

| Method | #Params | Caltech101 | OxfordPets | Flowers102 | Food101 | FGVCAircraft | SUN397 | DTD | SVHN | EuroSAT | Resisc45 | CLEVR | UCF101 | ImageNet | Average |
|---|---|---|---|---|---|---|---|---|---|---|---|---|---|---|---|
| BAR | 37.6k | **85.8** | **82.2** | **68.0** | **58.3** | 10.7 | 23.3 | 47.2 | 15.2 | 39.6 | 23.6 | 26.1 | 23.3 | **54.9** | 42.9 |
| BLACKVIP | 9.9k | 81.6 | 78.9 | 56.6 | 47.8 | 9.6 | **46.3** | 41.1 | 30.7 | 22.5 | 37.0 | 26.3 | 37.0 | 49.4 | 43.4 |
| BPTVLM | 4.0k | 63.9 | 71.7 | 47.7 | 37.2 | 9.9 | 35.5 | 45.6 | 14.7 | 44.8 | 37.0 | 24.8 | 35.5 | 39.5 | 39.1 |
| **ZIP** | 0.4k | 72.1 | 82.0 | 59.3 | 40.9 | **11.8** | 45.3 | **49.4** | **34.2** | 48.7 | 44.3 | 31.4 | 40.6 | 49.7 | **46.9** |

Table 8: Base-to-new generalization performance. H represents the harmonic mean, providing a balanced measure of accuracy across seen and unseen classes (Xian et al., 2017). Feature sharing consistently surpasses unshared models in base, new, and harmonic mean evaluations, demonstrating its effectiveness in improving generalization to novel classes.

| Method | Set | Caltech101 | OxfordPets | Flowers102 | Food101 | FGVCAircraft | SUN397 | DTD | SVHN | EuroSAT | Resisc45 | CLEVR | UCF101 | ImageNet | Average |
|---|---|---|---|---|---|---|---|---|---|---|---|---|---|---|---|
| Unshared | Base | 96.3 | 94.5 | 68.6 | **89.9** | 29.4 | 67.9 | 56.6 | 51.9 | 81.2 | 77.4 | 43.2 | 72.5 | 71.0 | 69.3 |
| Shared | | 96.6 | 94.9 | 72.1 | 89.9 | 29.8 | 70.3 | 61.7 | 52.9 | 84.0 | 81.6 | 50.1 | 75.1 | 72.1 | 71.6 |
| Unshared | New | **93.9** | 93.8 | 71.5 | 89.5 | 31.3 | 70.5 | 47.1 | **46.1** | 66.1 | 60.1 | 25.0 | 67.4 | 64.5 | 63.6 |
| Shared | | 93.2 | 97.0 | 73.4 | 90.0 | 32.0 | 71.5 | 51.0 | 45.8 | 64.4 | 65.2 | 26.8 | 69.5 | 65.6 | 65.0 |
| Unshared | Harmonic | **95.1** | 94.1 | 70.0 | 89.7 | 30.3 | 69.2 | 51.4 | 48.8 | **72.9** | 67.7 | 31.7 | 69.9 | 67.6 | 66.0 |
| Shared | | 94.9 | 95.9 | 72.8 | 89.9 | 30.9 | 70.9 | 55.8 | 49.1 | 72.9 | 72.5 | 34.9 | 72.2 | 68.7 | 68.2 |

Table 9: Cross-dataset transfer and out-of-distribution generalization performance. After training on ImageNet (*i.e.*, source) with 16-shot data per class, ZIP is evaluated on 12 target datasets for CDT and 4 ImageNet variants for OOD. Feature sharing consistently enhances transferability and robustness, outperforming the unshared models across both CDT and OOD tasks, highlighting its value in adapting to diverse and unseen distributions.

| | Source | CDT Target | | | | | | | | | | | | | OOD Target | | | | |
|---|---|---|---|---|---|---|---|---|---|---|---|---|---|---|---|---|---|---|---|
| Method | ImageNet | Caltech101 | OxfordPets | Flowers102 | Food101 | FGVCAircraft | SUN397 | DTD | SVHN | EuroSAT | Resisc45 | CLEVR | UCF101 | Average | ImageNet-A | ImageNetV2 | ImageNet-R | ImageNet-Sketch | Average |
| Unshared | 65.2 | **91.3** | 85.4 | 64.2 | **84.2** | 19.6 | 58.3 | 38.3 | **27.9** | **46.3** | 53.3 | **17.6** | 61.3 | 54.0 | 47.2 | 59.0 | **75.2** | 45.0 | 56.6 |
| Shared | **66.0** | 90.4 | **85.6** | **65.6** | 83.6 | **20.5** | **60.6** | **40.9** | 27.0 | 42.3 | **55.6** | 14.5 | **63.6** | **54.2** | **47.8** | **59.5** | 74.7 | **45.4** | **56.9** |

## B.3 PERFORMANCE ON ALTERNATIVE MODEL FAMILIES.

We extend our evaluation to SigLIP (Zhai et al., 2023), a vision-language model distinct from CLIP, to assess the versatility of ZIP across different model families. The results, presented in Table 7, indicate that while ZIP does not achieve the best performance on all datasets, it records significantly higher accuracy on several tasks, including DTD, SVHN, EuroSAT, Resisc45, CLEVR, and UCF101. Notably, ZIP achieves the highest average accuracy across datasets, underscoring its robustness and adaptability.

These findings highlight the generality of our method, demonstrating that ZIP consistently delivers strong performance across diverse datasets and model architectures, outperforming existing BBPT methods in key scenarios. This reinforces the potential of ZIP as a reliable approach for black-box prompt tuning in varied vision-language models.

### B.4 IMPACT OF FEATURE SHARING ON GENERALIZATION PERFORMANCE

To further assess the impact of feature sharing on generalization performance, particularly in unseen domains, we conducted experiments with and without feature sharing. The results are presented in Table 8 and Table 9.

From these results, we observe that feature sharing significantly enhances generalization across various tasks. For base-to-new generalization, feature sharing yields substantial performance improvements, with average gains of +2.3%, +1.4%, and +2.2% across different datasets. In cross-dataset transfer and out-of-distribution generalization tasks, the improvements are more moderate, averaging +0.2% and +0.3%, respectively.

These findings highlight the effectiveness of feature sharing in improving generalization to unseen datasets, especially in tasks requiring robust representations across varying distributions. While the gains in OOD settings are smaller, they still indicate the potential benefits of this approach. Future work will focus on an in-depth analysis of the relationship between feature sharing and generalization, aiming to uncover the mechanisms driving these improvements. This could offer valuable insights for broader domain generalization research and practical applications.

### B.5 LOW-RANK APPROXIMATION WITH DIAGONAL MATRIX

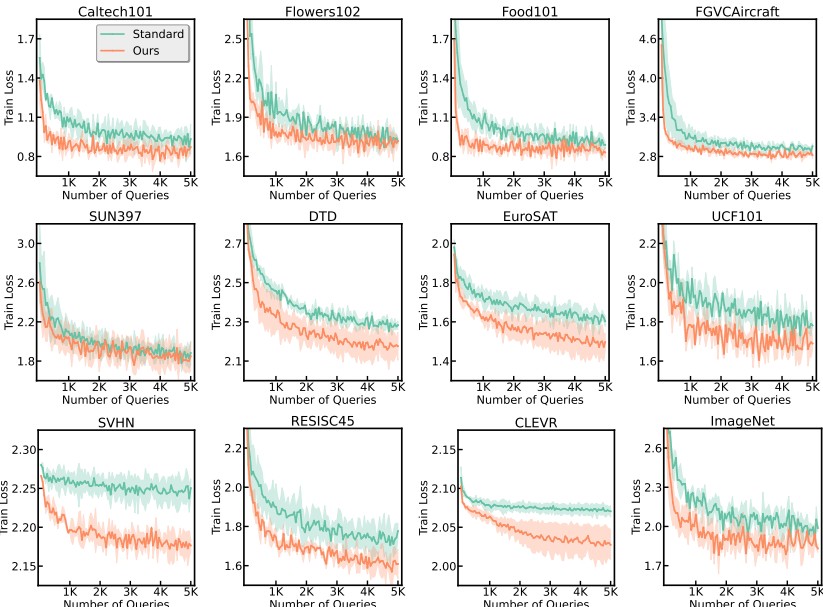

Figure 14: Effects of low-rank approximation with diagonal matrix across various vision-language tasks.

To further validate the effectiveness of our low-rank approximation with a diagonal matrix, introduced in Section 4.1, we conducted a comprehensive ablation study. This study compares the standard dimensionality reduction technique with our proposed low-rank approximation, evaluated in two settings.

First, we fixed the intrinsic dimensionality at 500 for both the standard method and our approach. However, our method applies an additional low-rank approximation with a diagonal matrix, reducing the parameter size to 417. As shown in Figure 14, this results in improved training speed.

Next, to isolate the effects of the low-rank approximation, we set the parameter size to 417 for both methods, demonstrating that hyper-parameter size alone is not the key factor driving the efficiency gains. As illustrated in Figure 15, our low-rank approximation method retains core information while reducing parameters, significantly enhancing both training speed and performance.

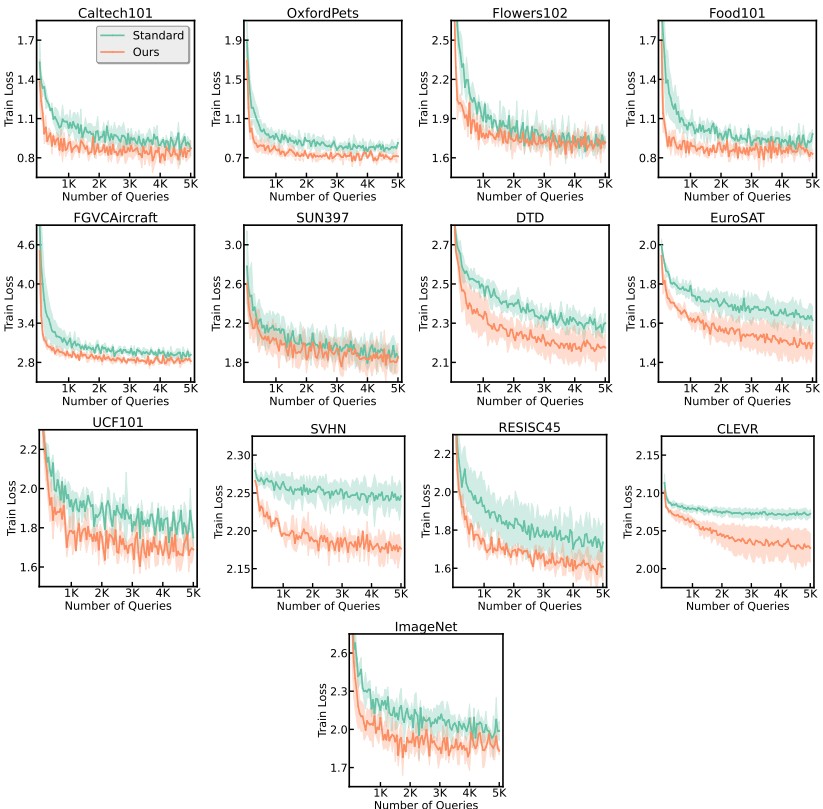

Figure 15: Effects of low-rank approximation with diagonal matrix at fixed parameter size (*i.e.*, 417) across various vision-language tasks.

Table 10: Benefits of low-rank approximation with diagonal matrix. Our method outperforms both standard dimensionality reduction and LoRA, showing significant improvements in test accuracy.

| Method | Caltech101 | OxfordPets | Flowers102 | Food101 | FGVCAircraft | SUN397 | DTD | SVHN | EuroSAT | Resisc45 | CLEVR | UCF101 | ImageNet | Average |
|---|---|---|---|---|---|---|---|---|---|---|---|---|---|---|
| Standard | 90.9 | 88.1 | 67.5 | 84.6 | 23.8 | 57.9 | 43.2 | 31.5 | 56.5 | 58.3 | 18.3 | 65.3 | 62.3 | 57.6 |
| LoRA | 90.7 | 89.3 | 68.1 | 85.0 | 23.7 | 57.4 | 43.9 | 36.0 | 59.2 | 57.0 | 21.2 | 65.2 | 62.6 | 58.4 |
| LoRA + Diagonal | 93.1 | 90.8 | 67.1 | 86.0 | 25.2 | 59.0 | 44.4 | 40.9 | 60.6 | 63.3 | 20.2 | 67.4 | 64.8 | 60.2 |

Additionally, we compared our technique to the LoRA-style approximation (Hu et al., 2022). Our method, which introduces only $r$ parameters in the diagonal matrix, effectively captures essential information from the parameter space, boosting the expressive power of the model without significant parameter overhead. Table 10 presents the test accuracy comparison between our approach, the standard dimensionality reduction method, and LoRA. Our method consistently outperforms both alternatives, demonstrating the clear advantage of integrating a diagonal matrix with low-rank approximation. These findings highlight the effectiveness of our approach in preserving model expressiveness while optimizing parameter efficiency, making it a compelling solution for efficient model training.

## B.6 INTRINSIC-DIMENSIONAL CLIPPING

In Figure 16 and 17, we further investigate the impact of intrinsic-dimensional clipping and the effect of varying the optimal clipping threshold across multiple datasets. The results indicate that applying intrinsic-dimensional clipping consistently enhances training accuracy and reduces loss across most datasets, demonstrating its effectiveness in stabilizing the training process.

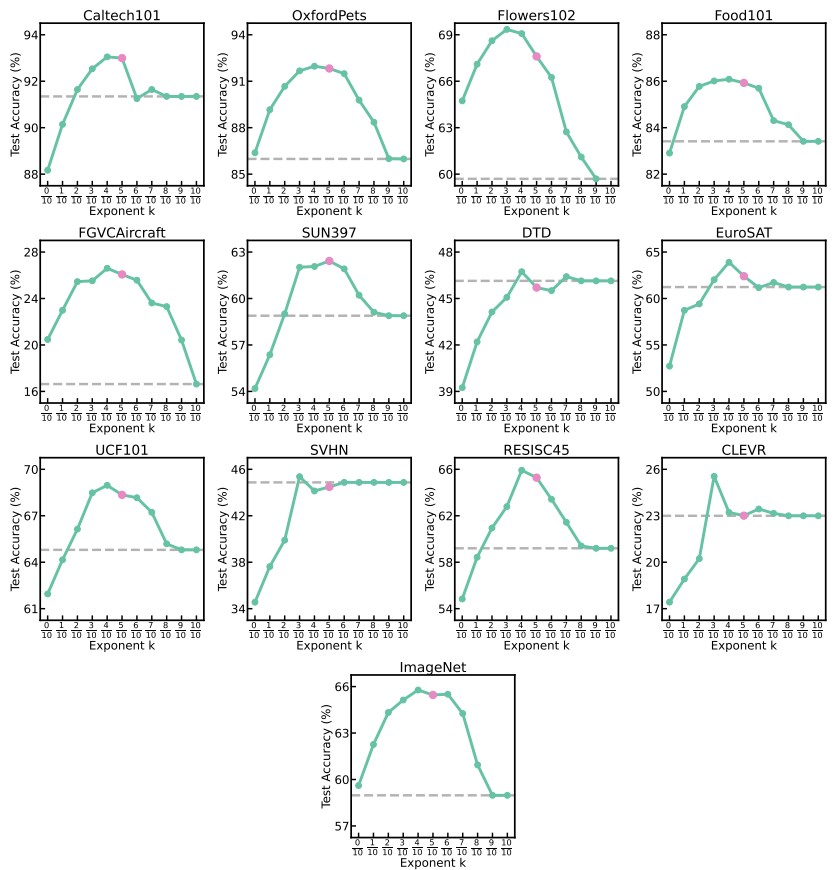

Figure 16: Effects of optimal threshold across various vision-language tasks.

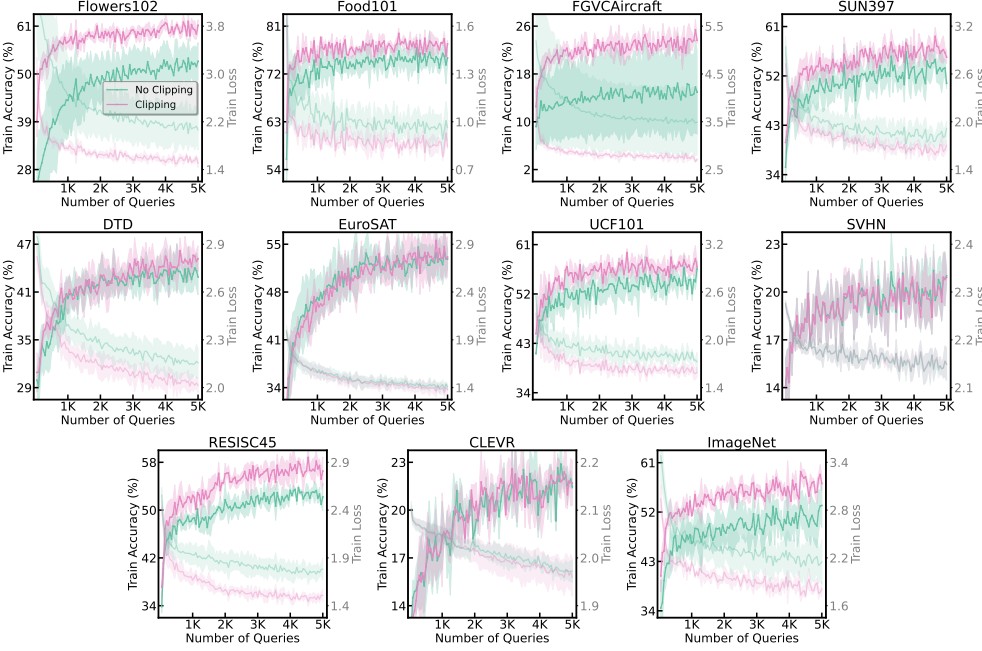

Figure 17: Effects of intrinsic-dimensional clipping across various vision-language tasks.

Table 11: Few-shot performance on 13 vision-language tasks with varying combinations of the proposed modules (*e.g.*, diagonal matrix, feature sharing (FS), and intrinsic-dimensional clipping). All the results are based on 16-shots per class. The **bold numbers** denote the highest accuracy of all baselines on each dataset, and the underlined values indicate the second.

| Number | Diagonal | FS | Clipping | Caltech101 | OxfordPets | Flowers102 | Food101 | FGVCAircraft | SUN397 | DTD | SVHN | EuroSAT | Resisc45 | CLEVR | UCF101 | ImageNet | **Average** |
|---|---|---|---|---|---|---|---|---|---|---|---|---|---|---|---|---|---|
| 1 | ✓ | ✗ | ✗ | 91.2 | 82.3 | 56.9 | 83.4 | 13.2 | 56.8 | 41.0 | 38.9 | 58.8 | 58.5 | 23.1 | 64.4 | 61.5 | 56.2 |
| 2 | ✗ | ✓ | ✗ | 90.1 | 89.3 | 65.3 | 84.6 | 22.7 | 60.6 | 42.4 | 38.4 | 59.3 | 59.8 | 18.9 | 66.7 | 63.4 | 58.6 |
| 3 | ✗ | ✗ | ✓ | 90.7 | 89.3 | 68.1 | 85.0 | 23.7 | 57.4 | 43.9 | 36.0 | 59.2 | 57.1 | 21.2 | 65.2 | 62.6 | 58.4 |
| 4 | ✓ | ✓ | ✗ | 91.3 | 86.0 | 59.7 | 83.4 | 16.6 | 58.9 | 46.1 | 44.9 | 61.2 | 59.2 | 23.0 | 64.8 | 59.0 | 58.0 |
| 5 | ✗ | ✓ | ✓ | 89.8 | 89.5 | 66.4 | 85.3 | 25.1 | 58.5 | 44.7 | 38.3 | 61.0 | 58.9 | 18.9 | 65.9 | 63.4 | 58.9 |
| 6 | ✓ | ✗ | ✓ | 93.1 | 90.8 | 67.1 | 86.0 | 25.2 | 59.0 | 44.4 | 40.9 | 60.6 | 63.3 | 20.2 | 67.4 | 64.8 | 60.2 |
| 7 | ✓ | ✓ | ✓ | 93.4 | 91.7 | 70.0 | 86.3 | 26.6 | 62.2 | 47.8 | 44.2 | 64.2 | 65.2 | 25.1 | 69.8 | 66.0 | 62.5 |

When evaluating test accuracy with varying gradient clipping thresholds, ZIP achieves near-optimal performance across the majority of datasets, consistently outperforming cases where no gradient clipping is applied. Although there are some exceptions, such as SVHN, DTD, and CLEVR, where gradient clipping does not yield significant improvements in test accuracy, the results remain comparable to ZIP without clipping, indicating that the technique does not hinder performance in these cases.

These findings substantiate that our intrinsic-dimensional clipping approach significantly improves the overall performance of zeroth-order optimization, and the selected $\sqrt{\delta}$ threshold effectively serves as a reliable and practical choice for enhancing training efficiency.

## B.7 ANALYSIS OF MODULE COMBINATIONS

We evaluate all combinations of the proposed modules, including diagonal matrix, feature sharing (FS), and intrinsic-dimensional clipping. The results are presented in Table 11. First, we observe that using all the proposed modules together results in significantly better performance compared to using individual modules or pairs of modules. This demonstrates that each component works harmoniously to contribute to the generation of effective results. Additionally, from the transitions $1 \rightarrow 6$, $4 \rightarrow 7$ and $5 \rightarrow 7$, we find that combining the low-rank approximation with diagonal matrix with intrinsic dimensional clipping yields more pronounced performance improvements (+4%, +4.5%, +3.6%) compared to other combinations. These findings suggest that while each component is effective on its own, their combination creates a complementary synergy that maximizes overall performance. In future work, we plan to conduct an in-depth analysis to uncover the underlying mechanisms behind this synergy. This will provide deeper insights into its practical utility, paving the way for its application to a broader range of tasks.

## B.8 SIGNIFICANCE TEST OF ZIP

To ensure the robustness of our results, we conduct a statistical significance analysis comparing ZIP and competing methods. While the average CDT accuracies of ZIP and the second-best method, BLACKVIP, appear close (Table 3), we extend the evaluation using 10 random seeds (1–10) to provide more reliable conclusions. A t-test was then perform to compute p-values for statistical significance. The results reveal that ZIP demonstrates statistically significant improvements in OOD tasks, while its performance in CDT tasks is comparable to BLACKVIP. Detailed results are presented in Table 12 and Table 13.

In CDT tasks, ZIP achieves statistically significant improvements on specific datasets such as Flowers ($p = 0.009$), Food101 ($p = 0.008$), SVHN ($p = 0.018$), and EuroSAT ($p = 0.003$). Conversely, BLACKVIP shows significantly higher performance on Aircraft ($p = 6.45e - 08$), DTD ($p = 3.87e - 04$), CLEVR ($p = 0.023$), and UCF101 ($p = 0.016$). These findings underscore that while ZIP delivers strong and consistent performance, dataset-specific characteristics can influence the effectiveness of each method.

In OOD tasks, ZIP consistently outperforms BLACKVIP in average performance (ZIP: 56.82%, BLACKVIP: 54.47%). Statistically significant improvements are observed on ImageNet-A ($p =$

Table 12: Cross-dataset transfer performance comparison between ZIP and BLACKVIP, including significance test results. The p-values from t-tests highlight statistically significant differences on specific datasets. ZIP demonstrates notable improvements in query efficiency and accuracy on datasets like Flowers and EuroSAT, while BLACKVIP excels in others such as Aircraft and DTD.

| Method | Source | CDT Target | | | | | | | | | | | | |
| | ImageNet | Caltech101 | OxfordPets | Flowers102 | Food101 | FGVCAircraft | SUN397 | DTD | SVHN | EuroSAT | Resisc45 | CLEVR | UCFI01 | Average |
|---|---|---|---|---|---|---|---|---|---|---|---|---|---|---|
| BLACKVIP | 65.19 | 92.69 | 86.28 | 64.72 | 83.36 | 22.31 | 62.04 | 42.88 | 17.68 | 39.71 | 55.98 | 15.70 | 64.07 | 54.82 |
| ZIP | 65.91 | 91.69 | 85.74 | 65.64 | 84.54 | 20.67 | 60.10 | 39.39 | 21.42 | 44.43 | 54.52 | 14.39 | 61.99 | 54.65 |
| t-stats | 1.745 | -1.879 | -0.690 | 2.942 | 2.974 | -8.771 | -2.755 | -4.349 | 2.615 | 3.409 | -1.588 | -2.479 | -2.650 | |
| p-value | 0.098 | 0.076 | 0.498 | 0.009 | 0.008 | 6.45e-08 | 0.013 | 3.87e-04 | 0.018 | 0.003 | 0.130 | 0.023 | 0.016 | |

Table 13: Out-of-distribution (OOD) generalization performance comparison between ZIP and BLACKVIP, including significance test results. The p-values from t-tests reveal statistically significant improvements by ZIP on datasets such as ImageNet-A, ImageNet-R, and ImageNet-Sketch, demonstrating its robustness under domain shifts. While ZIP achieves higher average performance, the results on ImageNetV2 show comparable performance with no statistical significance.

| Method | Source | OOD Target | | | | |
| | ImageNet | ImageNet-A | ImageNetV2 | ImageNet-R | ImageNet-Sketch | Average |
|---|---|---|---|---|---|---|
| BLACKVIP | 65.19 | 41.90 | 58.85 | 72.81 | 44.32 | 54.47 |
| ZIP | 65.91 | 47.94 | 59.48 | 74.64 | 45.25 | 56.82 |
| t-stats | 1.745 | 10.299 | 1.549 | 4.359 | 2.619 | |
| p-value | 0.098 | 5.66e-09 | 0.139 | 3.78e-4 | 0.017 | |

$5.66e - 09$), ImageNet-R ($p = 3.78e - 4$), and ImageNet-Sketch ($p = 0.017$). For ImageNetV2, the performance difference ($p = 0.139$) is not statistically significant, suggesting similar performance on this dataset.

The strong generalization performance of BLACKVIP stems from its image-dependent prompting strategy, drawing from prior works like CoCoOp (Zhou et al., 2022a). This design is tailored to improve generalization capabilities. In contrast, ZIP focuses on query efficiency, making it particularly effective in scenarios with limited API budgets. Despite differing objectives, ZIP demonstrates superior performance in OOD and base-to-new generalization tasks, while maintaining competitive performance in CDT tasks. These results highlight the balanced capabilities and adaptability of ZIP across various generalization settings.

## B.9 VALIDATION ACCURACY

To evaluate the generalization capability of ZIP and verify that it does not overfit, we report validation accuracies corresponding to the training results presented in Figure 2, 4, 6, and 17. These are illustrated in Figure 18, 19, 20, and Figure 21.

Across all datasets and experimental settings, the validation accuracy trends closely mirror the training accuracy, demonstrating consistent performance and negligible signs of overfitting. Notably, for zeroth-order optimization (Figure 20), the validation results confirm the stability and efficiency of the query-efficient design of ZIP. Similarly, the validation outcomes for intrinsic-dimensional clipping (Figure 21) highlight the effectiveness of the $\sqrt{\delta}$ threshold in enhancing both training and validation accuracy without introducing instability.

These findings provide strong evidence of robustness of ZIP, showing that it maintains reliable performance across both training and validation datasets. This underscores its generalization ability and query efficiency, further validating its applicability across diverse vision-language tasks, even under resource constraints.

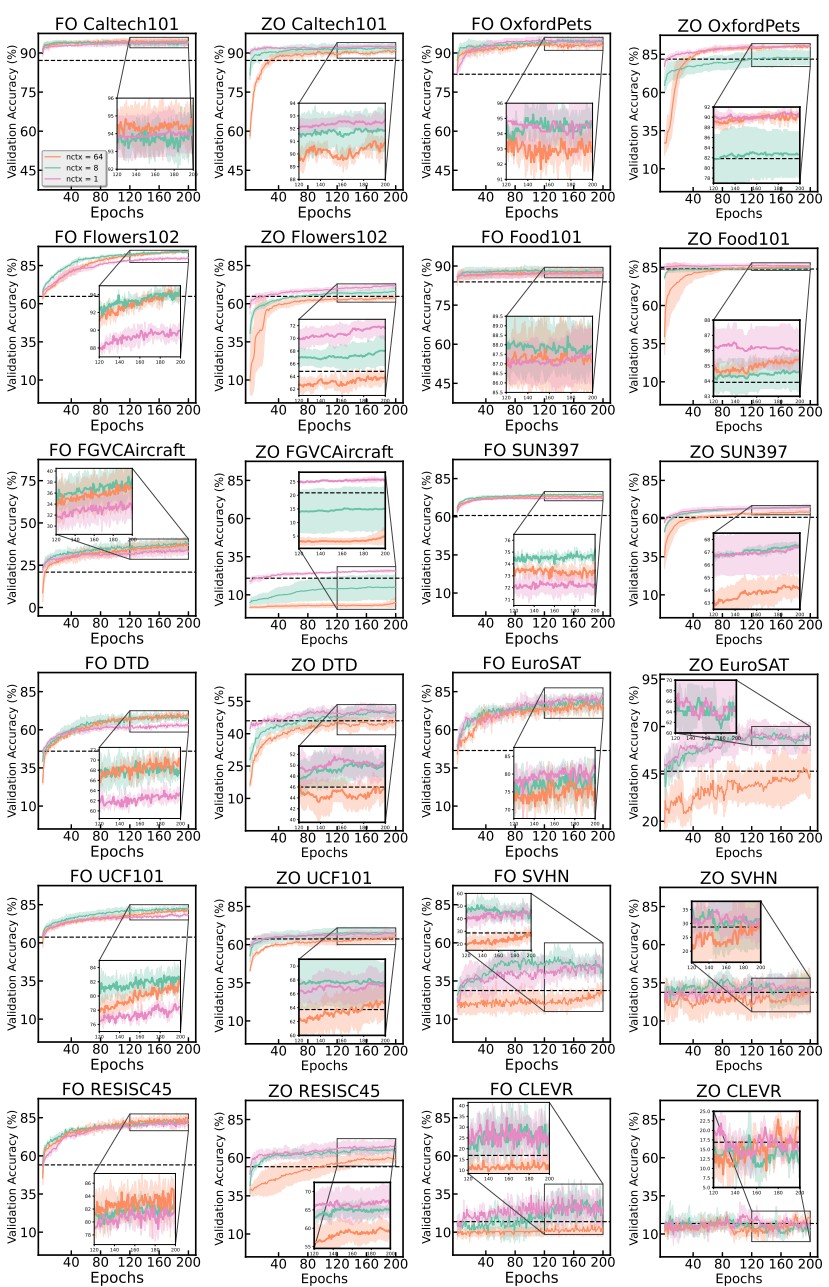

Figure 18: Validation curves illustrating the performance of different optimization methods across various vision-language tasks. The black dotted line represents no trainable parameters ($m = 0$, *i.e.*, only the [CLASS] token), serving as a baseline for comparison.

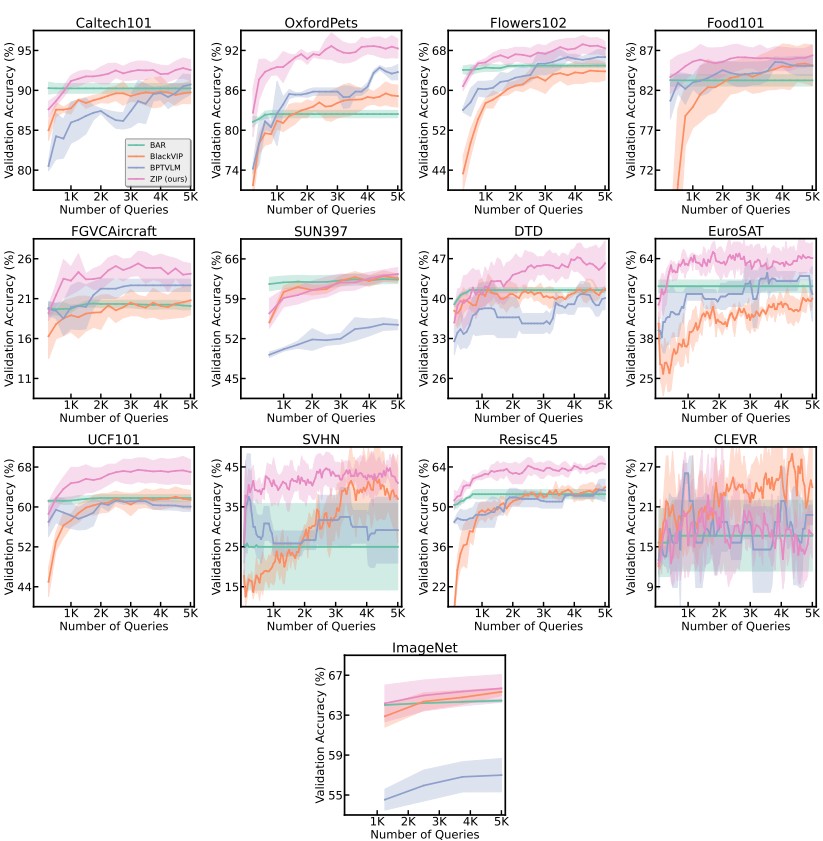

Figure 19: Validation curves under a 5,000 query budget across various vision-language tasks, showcasing ZIP's efficient utilization of limited queries to achieve competitive performance. The results demonstrate consistent validation trends, reflecting ZIP's robustness and generalization capabilities.

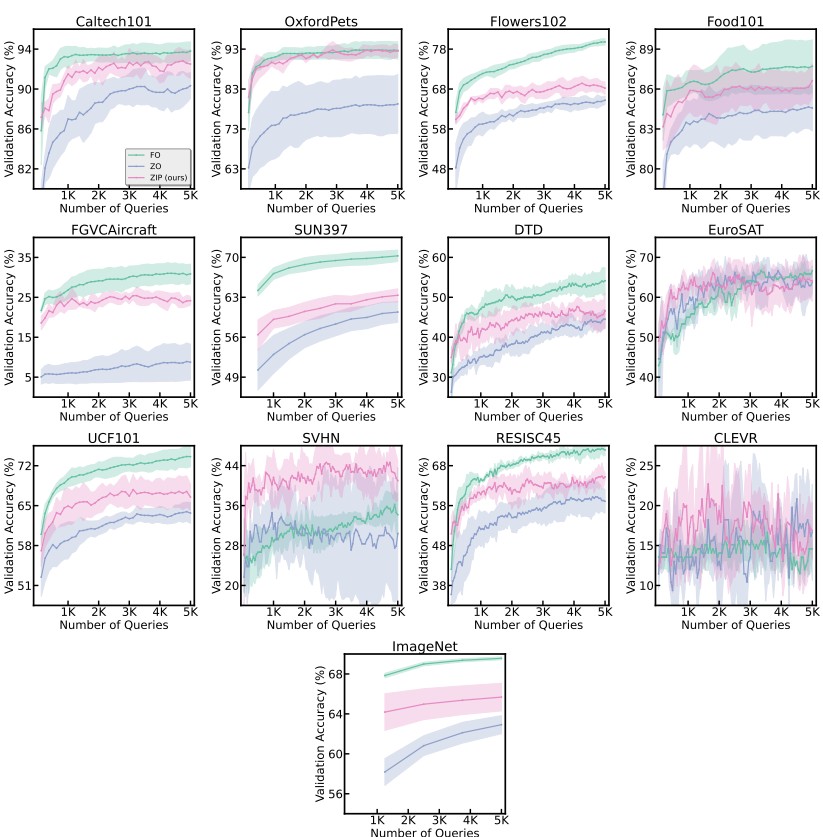

Figure 20: Validation curves comparing first-order, zeroth-order, and ZIP optimization methods across various vision-language tasks. ZIP bridges the gap between first-order and zeroth-order optimization, maintaining stable validation accuracy while achieving better generalization compared to standard zeroth-order methods.

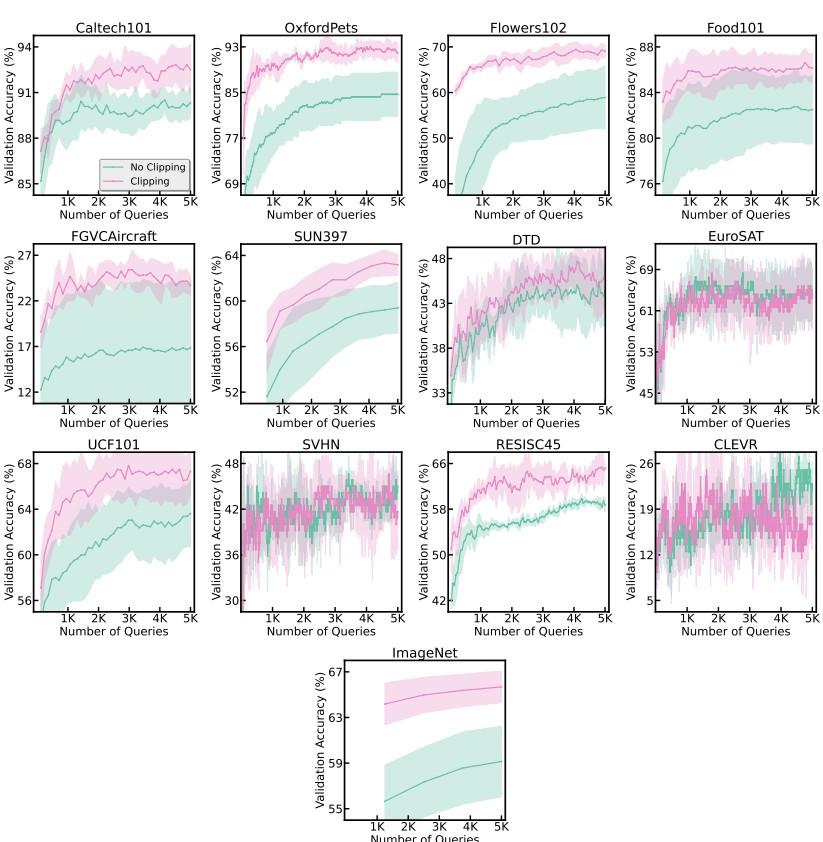

Figure 21: Validation curves illustrating the impact of zeroth-order gradient clipping on various vision-language tasks. The results emphasize the effectiveness of the $\sqrt{\delta}$ threshold in stabilizing training and improving validation accuracy, confirming its utility in zeroth-order optimization scenarios.

## C   EXPERIMENT DETAILS

### C.1   ALGORITHM

---

**Algorithm 1:** The training process of ZIP.

---

**Input:** The training data $\mathcal{D} = \{\boldsymbol{x}_i, \boldsymbol{y}_i\}$, pre-trained CLIP model $g$, projection matrix $\{\mathbf{M}_i\}_{i=1}^{m}$, learnable parameters of each context token $\theta_i$, gradient clipping threshold $\sqrt{\delta}$, context token counts $m$, number of gradient estimates $N$ for $N$-SPSA, smoothing parameter $c$, batch size $\mathcal{B}$, and API call budget $\mathcal{T}$.

**Function** $f(\Xi_t; X)$**:**

  Calculate the original token parameters $\theta_t$

  **for** $i$ to $m$ **do**

    $\theta_{t,i} = \theta_{0,i} + \mathbf{M}_i \boldsymbol{w}_{t,i}$

  **end**

  Forward propagate through CLIP model with reconstructed tokens $\widetilde{g} = g(\theta_t; X)$

  **return** $\widetilde{g}$

**Function** $N\text{-SPSA}(\Xi_t, c, N, X)$**:**

  **for** $n$ to $N$ **do**

    Sample $a \sim Uniform(0,1)$, with ensuring $a$ is not 0

    Sample $z_n \sim Bernoulli(a : 0.5, -a : 0.5)$

    Calculate the first loss $f(\Xi_t + c z_n; X)$

    Calculate the second loss $f(\Xi_t - c z_n; X)$

    Calculate the $n$-th gradient estimation

    $\widehat{\nabla} f_n(\Xi_t; X) = \frac{f(\Xi_t + c z_n; X) - f(\Xi_t - c z_n; X)}{2c}(z_n)^{-1}.$

  **end**

  Calculate $N$-SPSA gradient estimation $\widehat{\nabla} f(\Xi_t; X) = \frac{1}{N} \sum_{n=1}^{N} \widehat{\nabla} f_n(\Xi_t; X)$

  **return** $\widehat{\nabla} f(\Xi_t; X)$

Initialize $\Xi_0, \mathbf{U}_0, \boldsymbol{s}_0, \mathbf{V}_0$

**for** $t$ to $\mathcal{T}/2N$ **do**

  **for** each training mini-batch $X, Y$ **do**

    Calculate the weight matrix $\Xi_t = [\boldsymbol{w}_{t,1} | \boldsymbol{w}_{t,2} | \cdots | \boldsymbol{w}_{t,q}] = \mathbf{U}_t \text{diag}(\boldsymbol{s}_t) \mathbf{V}_t^T + \boldsymbol{u}_t \otimes \mathbb{1}$

    Calculate the gradient estimation $\widehat{\nabla} f(\Xi_t; X)$ using $N\text{-SPSA}(\Xi_t, c, N, X)$

    Calculate the clipping coefficient $\alpha_t = \min\left(\frac{\sqrt{\delta}}{\sqrt{\sum_{i=1}^{\delta} \widehat{\nabla} f(\theta_t)_i^2}}, 1\right)$

    Gradient descent using clipping $\Xi_{t+1} = \Xi_t - \eta_t \alpha_t \widehat{\nabla} f(\Xi_t)$

  **end**

**end**

---

During the training process, our method, ZIP, initiates by calculating the low-rank approximation and integrating shared feature representations. These approximations are subsequently utilized to reconstruct the original parameter space through random projection, allowing ZIP to generate the prompt representations necessary for loss computation efficiently. To ensure clarity and provide a comprehensive understanding of the training procedure, the summarized training algorithm can be found in Algorithm 1, which outlines each stage of the process for easy reference.

### C.2   COMPARISON OF BLACK-BOX SETTINGS IN VLMS

Our work builds upon the Language-model-as-a-Service (LMaaS) framework (Sun et al., 2022b), which envisions large language models as services accessible via APIs. This paradigm has attracted significant attention in previous studies as a practical and flexible setting for black-box prompt tuning (Sun et al., 2022b; Yu et al., 2023). In the LMaaS framework, models are treated as opaque systems, and users fine-tune them through external signals—such as logits or losses—without requiring direct access to the internal parameters. Moreover, the LMaaS framework accepts soft prompt inputs, further enhancing its adaptability. Although current models like ChatGPT (OpenAI, 2023) and Gemini (Google, 2023) do not yet support this specific configuration, the LMaaS approach

| | Black-box Assumption | |
|---|---|---|
| **Method** | Access Permission | Prompt Type |
| BAR (Tsai et al., 2020) | Loss | Hard |
| BLACKVIP (Oh et al., 2023) | Loss | Hard |
| BPTVLM (Yu et al., 2023) | Loss | Soft |
| LFA (Ouali et al., 2023) | Logits | Hard |
| CraFT (Wang et al., 2024) | Logits | Soft |
| ZIP (ours) | Loss | Soft |

Table 14: Comparison of black-box assumptions. We compare our setting with prior methods in terms of access permission and prompt type.

establishes a foundation for future methods that emphasize flexible, efficient, and secure fine-tuning of black-box models.

As detailed in Table 14, our approach adheres to a black-box setting where only loss values are accessible while soft prompts are utilized. This design is particularly advantageous as it avoids the need for accessing logits, *i.e.*, a type of sensitive information that should ideally remain concealed for security and privacy reasons. In contrast, methods such as LFA (Ouali et al., 2023) and CraFT (Wang et al., 2024) rely on logits, which raises concerns over model confidentiality and may not be feasible in many real-world API scenarios. By leveraging loss-based feedback, our method conforms more closely to practical constraints and fosters the development of robust optimization strategies capable of deriving meaningful insights from limited information.

## C.3 DATASET DETAILS

| | | | | *Classification Tasks* | |
|---|---|---|---|---|---|
| **Dataset** | **#Train** | **#Valid** | **#Test** | **Classification Type** | **Manual Prompt** |
| ImageNet | 1.28M | N/A | 50,000 | Generic object | "a photo of a [CLASS]." |
| Caltech101 | 4,128 | 1,649 | 2,465 | Generic object | "a photo of a [CLASS]." |
| OxfordPets | 2,944 | 736 | 3,669 | Fine-grained objects | "a photo of a [CLASS], a type of pet." |
| Flowers102 | 4,093 | 1,633 | 2,463 | Fine-grained objects | "a photo of a [CLASS], a type of flower." |
| Food101 | 50,500 | 20,200 | 30,300 | Fine-grained objects | "a photo of [CLASS], a type of food." |
| FGVCAircraft | 3,334 | 3,333 | 3,333 | Fine-grained objects | "a photo of a [CLASS], a type of aircraft." |
| SUN397 | 15,880 | 3,970 | 19,850 | Scene | "a photo of a [CLASS]." |
| DTD | 2,820 | 1,128 | 1,692 | Text | "[CLASS] texture." |
| SVHN | 73,257 | 26,032 | 26,032 | Digit | "This is a photo of a [CLASS]." |
| EuroSAT | 13,500 | 5,400 | 8,100 | Satellite | "a centered satellite photo of a [CLASS]." |
| Resisc45 | 6,300 | 2,520 | 7,560 | Scene | "This is a photo of a [CLASS]." |
| CLEVR | 70,000 | 15,000 | 15,000 | Diagnosis | "This is a photo of [CLASS] objects." |
| UCF101 | 7,639 | 1,898 | 3,783 | Action | "a photo of a person doing [CLASS]." |
| ImageNetV2 | N/A | N/A | 10,000 | Generic object | "a photo of a [CLASS]." |
| ImageNet-Sketch | N/A | N/A | 50,889 | Sketch image | "a photo of a [CLASS]." |
| ImageNet-A | N/A | N/A | 7,500 | Adversarially filtered image | "a photo of a [CLASS]." |
| ImageNet-R | N/A | N/A | 30,000 | Cartoon, Sculptures, Paintings | "a photo of a [CLASS]." |

Table 15: The datasets used in this study, along with the corresponding manual prompts. Samples are drawn exclusively from the original training set to ensure consistency with baseline data.

In this study, we leverage a total of 13 general classification datasets and 4 out-of-distribution (OOD) datasets, widely used in prior research. These 13 classification tasks are employed to comprehensively evaluate the performance of ZIP in general few-shot learning, base-to-new generalization, and cross-dataset transfer scenarios. Additionally, the 4 OOD datasets are used to rigorously assess the ability of ZIP to handle out-of-distribution generalization. A detailed overview of each dataset, including task descriptions and evaluation metrics, is provided in Table 15.

## C.4 HYPER-PARAMETERS

To achieve stable and accurate gradient approximations, zeroth-order optimization algorithms typically perform multiple gradient estimations, with the results being averaged to obtain a more reliable

| Hyper-parameter | Assignment | Method |
|---|---|---|
| initial LR | {40.0, 20.0, 10.0, 5.0, 1.0} | BAR |
| initial LR ($a_1$) | {1.0, 0.1, 0.01, 0.005} | BLACKVIP, ZIP |
| min LR | {0.1, 0.01, 0.001} | BAR |
| decaying step | {0.9, 0.5, 0.1} | BAR |
| LR decaying factor | {0.6, 0.5, 0.4, 0.3} | BLACKVIP, ZIP |
| initial PM ($c_1$) | {0.01, 0.005, 0.001} | BLACKVIP, ZIP |
| PM decaying factor | {0.2, 0.1} | BLACKVIP, ZIP |
| std. of perturbation | {1.0, 0.5} | BAR |
| smoothing | {0.1, 0.01, 0.001} | BAR |
| gradient smoothing | {0.9, 0.7, 0.5, 0.3} | BLACKVIP |
| population size | {5, 10, 15, 20} | BPTVLM |
| intrinsic dimensionality | {500, 1000, 2000} | BPTVLM, ZIP |
| rank | {1, 3, 5} | ZIP |
| visual tokens | {5, 10} | BPTVLM |
| text tokens | {5, 10} | BPTVLM |

Table 16: Hyper-parameter search range for BBPT approaches.

gradient estimate. Following the methodology outlined in Oh et al. (2023), we repeat this gradient estimation process five times for all zeroth-order-based baselines to ensure consistency and robustness. For SPSA methods, we tune key hyper-parameters, including the perturbation magnitude and decay factor. For evolutionary strategies, we adjust the population size, intrinsic dimensionality, and the number of visual and text tokens. The search ranges for these hyper-parameters are based on the recommendations provided by the authors of BAR (Tsai et al., 2020), BLACKVIP (Oh et al., 2023), and BPTVLM (Yu et al., 2023), and are summarized in Table 16. Regarding the learning objectives, cross-entropy loss is employed for BLACKVIP and BPTVLM, while focal loss is used for BAR. All BBPT experiments utilize a batch size of 128 across all datasets, ensuring consistent and comparable evaluation.

## C.5 BASELINE DETAILS

### C.5.1 ZERO-SHOT CLIP

CLIP (Radford et al., 2021) is a prominent vision-language foundation model widely employed across various tasks, such as classification, segmentation, and other vision-language applications. Trained on large-scale image-text datasets, CLIP has demonstrated exceptional effectiveness in numerous downstream tasks, thanks to its ability to leverage visual concepts learned from natural language supervision. It performs zero-shot classification using manually crafted prompt templates (*e.g.*, "a photo of a [CLASS]."). Due to its versatility and strong performance, CLIP serves as the backbone for many black-box prompt tuning models, including our proposed method, ZIP.

### C.5.2 BAR

Originally developed for transferring knowledge from an ImageNet pre-trained model to the medical domain, BAR (Tsai et al., 2020) reprograms pre-trained models using a frame-shaped, learnable program that embeds the target task image within this frame and optimizes it via zeroth-order algorithms. The size of this learnable program is adjusted based on the input image resolution. For example, in the original study, when the resolution of the downstream image was larger than that of the pre-trained model, an embedded target image size of $64 \times 64$ was used within a $299 \times 299$ learnable program. In contrast, BLACKVIP (Oh et al., 2023) modified this approach by designing an embedded image resolution of $194 \times 194$ to avoid performance degradation caused by the heavy-padding of thin images within the prompt. In this paper, we adopt the settings established by BLACKVIP (Oh et al.,

2023) when optimizing BAR, ensuring consistency and addressing the limitations of the original design.

### C.5.3   BLACKVIP

BLACKVIP (Oh et al., 2023) generates input-conditional visual prompts for each image via a projection network, allowing prompts to adapt dynamically to the specific features of each input. For the optimization process, BLACKVIP employs Simultaneous Perturbation Stochastic Approximation with Gradient Correction (SPSA-GC), which integrates Nesterov Accelerated Gradients (NAG) (Nesterov, 1983), enhancing the efficiency of zeroth-order training. Unlike other methods such as CoCoOp (Zhou et al., 2022a), which optimize additional input-attached parameters, BLACKVIP focuses exclusively on the projection network, effectively creating adaptive, input-conditioned visual prompts for BBPT tasks. While this design choice makes BLACKVIP highly adaptable and well-suited for black-box settings, the large number of parameters introduced by the projection networks can negatively impact training efficiency, posing a challenge in resource-constrained environments.

### C.5.4   BPTVLM

BPTVLM (Yu et al., 2023) utilizes evolutionary strategies for BBPT, distinguishing itself from previous approaches. In this method, BPTVLM introduces learnable parameters into both text and image prompts, enabling a more comprehensive adaptation to various tasks. To enhance efficiency, BPTVLM incorporates the concept of intrinsic dimensionality, reducing the overall number of learnable parameters by applying a random projection matrix to both text and image prompts. This approach effectively balances adaptability and parameter efficiency, making BPTVLM a more versatile option for BBPT scenarios.

