# OpenReview forum: "ZIP: An Efficient Zeroth-order Prompt Tuning for Black-box Vision-Language Models"
_ICLR.cc/2025/Conference — ICLR 2025 Poster_

### Official Review · Reviewer_NLRG · 2024-10-31

**Soundness:** 3
**Presentation:** 3
**Contribution:** 3
**Rating:** 6
**Confidence:** 3

**Summary:**

This paper introduces ZIP, a zeroth-order intrinsic-dimensional prompt-tuning method designed to efficiently optimize black-box vision-language models. By leveraging low-rank approximation, feature sharing, and intrinsic-dimensional gradient clipping, ZIP achieves faster training speeds and superior generalization performance while significantly reducing query requirements. Extensive experiments on diverse tasks demonstrate ZIP's robustness and query efficiency, outperforming existing BBPT methods and establishing it as a practical approach for resource-constrained scenarios.

**Strengths:**

1.The paper presents a novel black-box prompt-tuning method, effectively addressing the issue in zeroth-order methods where an increase in trainable parameters adversely impacts accuracy. By reducing the number of parameters and query requirements, the proposed approach is well-suited for practical applications with limited query budgets.

2.The paper demonstrates strong performance across three extensive and diverse experimental settings, which effectively validate the method’s efficacy. The ablation studies further support the approach, particularly highlighting that the feature-sharing technique helps preserve the model’s expressive capacity.

3.The intrinsic-dimensional clipping mechanism in ZIP requires no manual hyperparameter tuning, making it highly practical and user-friendly.

4.The paper is well-written, with clear explanations and logical organization that make the proposed method and its contributions easy to understand.

**Weaknesses:**

1.Although the paper performs ablation studies on individual modules such as  low-rank approximation with a diagonal matrix and feature sharing, it lacks ablation experiments on different combinations of these modules.   Without evaluating different combinations, it is challenging to fully understand the synergistic effects and the relative contributions of each module to the overall performance.



2.The paper lacks an ablation study to isolate the effect of low-rank approximation alone, making it unclear if improvements are mainly due to the diagonal matrix. This analysis would clarify the diagonal matrix's contribution.

**Questions:**

Suggestions:

The caption for Figure 1 should include citations for the baseline methods (BAR, BlackVIP, BPT-VLM) to provide appropriate references and context for these comparisons. This would enhance clarity for readers unfamiliar with these specific methods.

**Details Of Ethics Concerns:**

No ethical concerns.

---

> ### Author Response · Authors · 2024-11-22
> **Response to Reviewer NLRG**
>
> We sincerely thank the reviewer for finding our work practical and effective, and giving us constructive feedback to improve further. While we respond to the reviewer’s specific comments as below, we would be keen to engage in any further discussion.
>
> ---
>
> **Ablation for different combinations of modules**
> > Although the paper performs ablation studies on individual modules such as low-rank approximation with a diagonal matrix and feature sharing, it lacks ablation experiments on different combinations of these modules. Without evaluating different combinations, it is challenging to fully understand the synergistic effects and the relative contributions of each module to the overall performance.
>
> Thank you for your suggestion. We have evaluated all possible combinations of {diagonal matrix, feature sharing (FS), intrinsic-dimensional clipping}. The results are provided below.
>
> | Number | Diagonal | FS | Clipping | Caltech | Pets | Flowers | Food | Aircraft | SUN | DTD  | SVHN | EuroSAT | Resisc | CLEVR | UCF | IN | Avg |
> |----|----|----|----|----|----|----|----|----|----|----|----|----|----|----|----|----|----|
> | 1 | ✓ | ✗ | ✗ | 91.2 | 82.3 | 56.9 | 83.4 | 13.2 | 56.8 | 41.0 | 38.9 | 58.8 | 58.5 | 23.1  | 64.4 | 61.5 | 56.2 |
> | 2 | ✗ | ✓ | ✗ | 90.1 | 89.3 | 65.3 | 84.6 | 22.7 | 60.6 | 42.4 | 38.4 | 59.3 | 59.8 | 18.9 | 66.7 | 63.4 | 58.6 |
> | 3 | ✗ | ✗ | ✓ | 90.7 | 89.3 | 68.1 | 85.0 | 23.7 | 57.4 | 43.9 | 36.0 | 59.2 | 57.1 | 21.2 | 65.2 | 62.6 | 58.4 |
> | 4 | ✓ | ✓ | ✗ | 91.3 | 86.0 | 59.7 | 83.4 | 16.6 | 58.9 | 46.1 | **44.9** | 61.2 | 59.2 | 23.0 | 64.8 | 59.0 | 58.0 |
> | 5 | ✗ | ✓ | ✓ | 89.8 | 89.5 | 66.4 | 85.3 | 25.1 | 58.5 | 44.7 | 38.3 | 61.0 | 58.9  | 18.9 | 65.9 | 63.4  | 58.9 |
> | 6 | ✓ | ✗ | ✓ | 93.1  | 90.8  | 67.1 | 86.0 | 25.2 | 59.0 | 44.4 | 40.9 | 60.6 | 63.3  | 20.2  | 67.4 | 64.8 | 60.2 |
> | 7 | ✓ | ✓ | ✓ | **93.4** | **91.7** | **70.0** | **86.3** | **26.6** | **62.2** | **47.8** | 44.2 | **64.2** | **65.2** | **25.1** | **69.8** | **66.0** | **62.5** |
>
> First, we observe that using all the proposed modules together results in significantly better performance compared to using individual modules or pairs of modules. This demonstrates that each component works harmoniously to contribute to the generation of effective results.
>
> Additionally, from the transitions 1&rarr;6, 4&rarr;7 and 5&rarr;7, we find that combining the low-rank approximation with diagonal matrix with intrinsic dimensional clipping yields more pronounced performance improvements (+4%, +4.5%, +3.6%) compared to other combinations.
>
> These findings suggest that while each component is effective on its own, their combination creates a complementary synergy that maximizes overall performance. In future work, we plan to conduct an in-depth analysis to uncover the underlying mechanisms behind this synergy. This will provide deeper insights into its practical utility, paving the way for its application to a broader range of tasks.
>
>
> ---
>
> **Effectiveness of diagonal matrix?**
> > The paper lacks an ablation study to isolate the effect of low-rank approximation alone, making it unclear if improvements are mainly due to the diagonal matrix. This analysis would clarify the diagonal matrix's contribution.
>
> We kindly remark that the ablation analysis the reviewer requests is already provided in Table 6 in Appendix. This results clearly shows that adding a simple diagonal matrix (# of parameters < 10) enhances the performance of low-rank approximation, improving the average performance by +1.8%.
>
> ---
>
> **Citation in Figure 1**
> > The caption for Figure 1 should include citations for the baseline methods (BAR, BlackVIP, BPT-VLM) to provide appropriate references and context for these comparisons. This would enhance clarity for readers unfamiliar with these specific methods.
>
> Thank you for your suggestion. We will reflect it to our final version.
>
> ---
>
> We sincerely appreciate the reviewer’s recognition of our work as “well-suited” and “user-friendly” for practical applications. We are also deeply grateful for the reviewer’s suggestion on additional ablations, which we believe has significantly improved our manuscript. We hope our response has adequately addressed your concerns, and yet, please let us know if there are any remaining issues. We would be eager to address them further.

---

> > ### Comment · Reviewer_NLRG · 2024-11-22
> >
> > Thank you for your detailed responses. To ensure these valuable additions are fully reflected, could you please integrate the new table, discussions, and citations into the revised manuscript? This will enhance the clarity and completeness of the final version.

---

> > > ### Author Response · Authors · 2024-11-25
> > > **Thank you for the response.**
> > >
> > > Thank you for your valuable feedback. As suggested, we have incorporated the new table and discussions into Appendix A and citations into Figure 1 of the revised manuscript. These additions will be integrated into the relevant sections of the main manuscript after the rebuttal process. Your guidance has been instrumental in enhancing the clarity and completeness of the final version. Please let us know if there are any other aspects you would like us to address.

---

### Official Review · Reviewer_Fqbz · 2024-11-01

**Soundness:** 3
**Presentation:** 4
**Contribution:** 3
**Rating:** 6
**Confidence:** 4

**Summary:**

The paper introduces ZIP, a zeroth-order prompt tuning method designed for efficient prompt optimization in black-box vision-language models, particularly under limited query budgets. ZIP achieves high efficiency by using low-rank representations and intrinsic-dimensional gradient clipping, which reduces query usage while maintaining robust performance. Evaluations on multiple benchmarks show that ZIP not only outperforms state-of-the-art methods in accuracy but also greatly enhances query efficiency.

**Strengths:**

(1) The paper is well-organized and accessible, with clear visuals and structured explanations that effectively communicate the method's strengths.

(2) ZIP innovatively enhances zeroth-order prompt tuning through intrinsic-dimensional gradient clipping and low-rank parameterization, making it highly efficient.

(3) Comprehensive evaluations demonstrate ZIP's superior accuracy and query efficiency across 13+ tasks, proving its practical value under query constraints.

**Weaknesses:**

(1) While ZIP outperforms existing BBPT methods, comparisons with additional baseline methods in zeroth-order optimization could strengthen claims of superiority.

(2) While ZIP shows strong performance on various tasks, its results on ImageNet in Table 1 are comparatively modest, suggesting limitations in scalability to complex datasets. An in-depth analysis of ZIP's performance on larger, diverse datasets would clarify its robustness and potential for broader application.

**Questions:**

(1) In Section 4.2, the paper introduces feature sharing to enhance expressiveness. Could the authors clarify whether this feature sharing technique affects the generalization ability on unseen datasets, and if so, how?

(2) ZIP has demonstrated strong results across vision-language tasks, but could the authors provide more insights into its potential for domain generalization? Specifically, how well does ZIP adapt to unseen domains or datasets outside the evaluated benchmarks, and would any adjustments be necessary to improve its robustness in such scenarios? Such as CoOp and CoCoOp.

(3)  Could the authors elaborate on the sensitivity of ZIP to the choice of intrinsic dimensionality and low-rank approximation parameters? How do these choices impact both performance and query efficiency?

---

> ### Author Response · Authors · 2024-11-24
> **Response to Reviewer Fqbz (part 1)**
>
> We sincerely appreciate the reviewer’s recognition and the constructive feedback. We have addressed the reviewer’s specific comments below, while we remain open to any additional suggestions.
>
> ---
>
> **Other ZO Optimization methods?**
> > Comparisons with additional baseline methods in zeroth-order optimization could strengthen claims of superiority.
>
> We kindly request the reviewer to clarify whether it is other BBPT methods (that leverage ZOO) or literally other zeroth-order methods (e.g., ZO-SGD, SPSA-GC [1]) that are being questioned. If it is the former, to the best of our knowledge, there are no additional BBPT methods leveraging ZOO beyond those already considered in our work. If it is the latter which involves comparisons between other ZOO algorithms and their clipped versions, such as SPSA-GC and SPSA-GC with clipping, please let us know. We will address this through additional experiments.
>
> ---
>
> **Complex datasets?**
> > Its results on ImageNet in Table 1 are comparatively modest, suggesting limitations in scalability to complex datasets. An in-depth analysis of ZIP's performance on larger, diverse datasets would clarify its robustness and potential for broader application.
>
> Thank you for raising this concern. We would like to point out that a 5,000 API query setting may not be sufficient to fully train a complex dataset such as ImageNet. For comparison, prior work [1] used significantly more queries, with 625,000 API queries employed for ImageNet training. To provide a more thorough evaluation of the performance of ZIP, we conduct additional experiments with 20,000 API queries. The results are summarized in the table below.
>
> | Method | Caltech | Pets | Flowers | Food | Aircraft | SUN | DTD  | SVHN | EuroSAT | Resisc | CLEVR | UCF | IN | Avg |
> |----|----|----|----|----|----|----|----|----|----|----|----|----|----|----|
> | BAR | 92.5 | 88.1 | 67.7 | 83.2 | 23.0 | 63.8 | 47.1 | 32.3 | 54.0 | 64.1 | 25.4 | 65.7 | 65.5 | 59.4 |
> | BlackVIP | 93.0 | 88.0 | 64.8 | 85.0 | 22.5 | 62.3 | 44.4 | 42.3 | 57.2 | 57.1 | **28.8** | 66.1 | 66.3 | 59.8 |
> | BPTVLM | 91.6 | 90.3 | 69.9 | 85.1 | 26.3 | 57.2 | 50.1 | 34.4 | 65.8 | 63.6 | 27.8 | 67.4 | 61.6 | 60.9 |
> | ZIP | **94.1** | **92.4** | **71.8** | **86.9** | **27.3** | **64.4** | **52.9** | **49.9** | **66.6** | **68.4** | 28.6 | **70.0**  | **67.2**  | **64.7**  |
>
> These findings reveal several key points. With a 5,000 API query budget, ZIP achieves an ImageNet accuracy of 66.0%, comparable to strong baselines such as Manual Prompt (66.7%) and BlackVIP (65.5%). Increasing the query budget to 20,000 further improves the accuracy of ZIP to 67.2%, outperforming those baselines by margins of 0.5% and 0.9%, respectively.
>
> Moreover, compared to prior work that leveraged significantly more API queries (e.g., BlackVIP: 67.1% on ImageNet), ZIP delivers outstanding performance using only 20,000 API queries. This demonstrates strong scalability and effectiveness, even on challenging datasets like ImageNet.
>
> If you still have any remaining concerns regarding this issue, we would sincerely appreciate it if you could recommend a complex dataset to evaluate ZIP. We will gladly consider it as part of our future work and look forward to sharing the results based on your suggestion.
>
> &nbsp;
>
> [1] Oh, Changdae, et al. “BlackVIP: Black-box visual prompting for robust transfer learning.”, CVPR, 2023.

---

> > ### Author Response · Authors · 2024-11-24
> > **Response to Reviewer Fqbz (part 2)**
> >
> > **Feature sharing and robustness**
> > > In Section 4.2, the paper introduces feature sharing to enhance expressiveness. Could the authors clarify whether this feature sharing technique affects the generalization ability on unseen datasets, and if so, how?
> >
> > Thank you for the question. We evaluate the generalization ability of feature sharing across unseen datasets. The results are summarized in the tables below.
> >
> > `Base-to-New Generalization`
> > | Method | Set | Caltech | Pets | Flowers | Food | Aircraft | SUN | DTD  | SVHN | EuroSAT | Resisc | CLEVR | UCF | IN | Avg |
> > |----|----|----|----|----|----|----|----|----|----|----|----|----|----|----|----|
> > | Unshared | Base | 96.3 | 94.5 | 68.6 | **89.9** | 29.4 | 67.9 | 56.6 | 51.9 | 81.2 | 77.4 | 43.2 | 72.5 | 71.0 | 69.3 |
> > | Shared | Base | **96.6** | **94.9** | **72.1**| **89.9** | **29.8** | **70.3** | **61.7** | **52.9** | **84.0** | **81.6** | **50.1** | **75.1** | **72.1** | **71.6** |
> > | Unshared | New | **93.9** | 93.8 | 71.5 | 89.5 | 31.3 | 70.5 | 47.1 | **46.1** | **66.1** | 60.1 | 25.0 | 67.4 | 64.5 | 63.6 |
> > | Shared | New | 93.2 | **97.0** | **73.4** | **90.0** | **32.0** | **71.5** | **51.0** | 45.8 | 64.4 | **65.2** | **26.8** | **69.5** | **65.6** | **65.0** |
> > | Unshared | Harmonic | **95.1** | 94.1 | 70.0 | 89.7 | 30.3 | 69.2 | 51.4 | 48.8 | **72.9** | 67.7 | 31.7 | 69.9 | 67.6 | 66.0 |
> > | Shared | Harmonic | 94.9 | **95.9** | **72.8** | **89.9** | **30.9** | **70.9** | **55.8** | **49.1** | **72.9** | **72.5** | **34.9** | **72.2** | **68.7** | **68.2** |
> >
> > `Cross Dataset Transfer`
> > | Method | IN | Caltech | Pets | Flowers | Food | Aircraft | SUN | DTD  | SVHN | EuroSAT | Resisc | CLEVR | UCF | Avg |
> > |----|----|----|----|----|----|----|----|----|----|----|----|----|----|----|
> > | Unshared | 65.2 | **91.3** | 85.4 | 64.2 | **84.2** | 19.6 | 58.3 | 38.3 | **27.9** | **46.3** | 53.3 | **17.6** | 61.3 | 54.0 |
> > | Shared | **66.0** | 90.4 | **85.6** | **65.6** | 83.6 | **20.5** | **60.6** | **40.9** | 27.0 | 42.3 | **55.6** | 14.5 | **63.6** | **54.2** |
> >
> > `Out-of-Distribution Generalization`
> > | Method | ImageNet-A | ImageNetV2 | ImageNet-R | ImageNet-Sketch | Average |
> > |----|----|----|----|----|----|
> > | Unshared | 47.2 | 59.0 | **75.2** | 45.0 | 56.6 |
> > | Shared | **47.8** | **59.5** | 74.7 | **45.4** | **56.9** |
> >
> > From these results, we observe that feature sharing positively impacts generalization across various tasks. In base-to-new generalization, it achieves significant performance improvements with average gains of +2.3%, +1.4%, and +2.2%. For cross-dataset transfer and out-of-distribution generalization, the improvements are moderate, with average gains of +0.2% and +0.3%, respectively.
> >
> > These findings suggest that feature sharing is effective for generalization on unseen dataset. In future work, we aim to conduct an in-depth analysis to better understand the relationship between feature sharing and generalization. This will provide valuable insights into its empirical utility and open new avenues for its application in broader domain generalization research.

---

> > > ### Author Response · Authors · 2024-11-24
> > > **Response to Reviewer Fqbz (part 3)**
> > >
> > > **Explanation of generalization**
> > > > ZIP has demonstrated strong results across vision-language tasks, but could the authors provide more insights into its potential for domain generalization? Specifically, how well does ZIP adapt to unseen domains or datasets outside the evaluated benchmarks,
> > >
> > > Thank you for the positive and constructive feedback. The strong generalization performance of ZIP can be attributed to two key factors:
> > >
> > > * **reduced model capacity**: Complex models are often prone to overfitting because they tend to memorize detailed properties of the training dataset rather than generalizing effectively. By reducing the capacity of the soft tokens through a series of low-rank approximations, we limit its ability to represent overly intricate patterns, thereby encouraging the model to focus on capturing broader, more generalizable features. This constraint on complexity directly mitigates the risk of overfitting, as the model is less likely to adapt excessively to task specific data during training.
> > > * **noisy gradient estimates in zeroth-order optimization**: Zeroth-order optimization methods, which rely on function evaluations rather than exact gradient computations, inherently produce noisy estimates of the gradients. This noise introduces stochasticity into the optimization process, acting as a form of regularization [1, 2, 3]. The presence of noise helps prevent the model from overfitting to outliers in the training data. Specifically, the noisy gradients make it less likely for the optimization algorithm to become finely tuned to the peculiarities of outlier data points, which might otherwise skew the learning process of the model. Moreover, this inherent noise encourages the model to focus on capturing general patterns within the data rather than memorizing specific instances. As a result, the model develops a more robust understanding of the underlying data distribution, enhancing its ability to generalize to unseen domains. The stochastic nature of the optimization helps the model explore a wider range of parameter configurations, increasing the likelihood of finding solutions that perform well across different datasets and reducing sensitivity to anomalies present in the training set.
> > >
> > > We speculate that these factors collectively hint at the potential of ZIP for achieving stable performance on unseen domains, as observed in our experiments, though more precise investigation is needed.
> > >
> > > ---
> > >
> > > **Adjustment to improve robustness?**
> > > > and would any adjustments be necessary to improve its robustness in such scenarios? Such as CoOp and CoCoOp.
> > >
> > > We appreciate the reviewer's question. CoCoOp adopts an input-conditional prompting scheme, improving (domain) generalization. However, this _adaptive_ prompting in fact requires the access to the image encoder of the vision-language model, which may not suit the setting of purely black-box models we consider in this work. Therefore, it is challenging for us to precisely forecast whether or not such a scheme would improve ZIP. One potential detour to realize this idea is leveraging an additional and separate image encoder, which being considered as an interesting extension, we will investigate further in future work.
> > >
> > > &nbsp;
> > >
> > > [1] Blanc, Guy, et al. “Implicit regularization for deep neural networks driven by an Ornstein-Uhlenbeck like process.” COLT, 2022.\
> > > [2] Damian, Alex, et al. “Label noise SGD provably prefers flat global minimizers.” NeurIPS, 2021.\
> > > [3] Ge, Rong, et al. “Escaping from saddle points - Online stochastic gradient for tensor decomposition.” COLT, 2015.

---

> > > > ### Author Response · Authors · 2024-11-24
> > > > **Response to Reviewer Fqbz (part 4)**
> > > >
> > > > **Sensitivity to intrinsic dimensionality and low-rank approximation**
> > > > > Could the authors elaborate on the sensitivity of ZIP to the choice of intrinsic dimensionality and low-rank approximation parameters? How do these choices impact both performance and query efficiency?
> > > >
> > > > Thank you for the question. We have evaluated the sensitivity of ZIP to intrinsic dimensionality and low-rank approximation parameters, with the results summarized in the tables below.
> > > >
> > > > `Dimensionality`
> > > > | Dim | Caltech | Pets | Flowers | Food | Aircraft | SUN | DTD  | SVHN | EuroSAT | Resisc | CLEVR | UCF | IN | Avg |
> > > > |----|----|----|----|----|----|----|----|----|----|----|----|----|----|----|
> > > > | 100 | 92.08 | 91.13 | 68.11 | 85.49 | 24.96 | 59.76 | 44.64 | 40.46 | 62.97 | 62.28 | 20.26 | 66.82 | 64.75 | 60.29 |
> > > > | 500 | **93.39** | 91.74 | 69.97 | 86.31 | 26.62 | 62.17 | 47.83 | 44.15 | **64.21** | 65.22 | 25.09 | **69.81** | 65.99 | **62.50** |
> > > > | 1000 | 93.23 | **92.14** | **70.50** | 85.36 | 26.83 | 62.92 | **47.85** | 44.15 | 62.47 | 64.78 | **26.31** | 69.04 | **66.47** | 62.47 |
> > > > | 2000 | 93.29 | 91.47 | 69.48 | **86.98** | **26.97** | **64.17** | 45.76 | **45.46** | 62.22 | **66.10** | 23.58 | 69.18 | 66.21 | 62.37 |
> > > >
> > > > Lower dimensionality improves query efficiency by simplifying the model, but it may reduce performance due to limited expressive power. On the other hand, higher dimensionality enhances expressive power and performance, but requires more API queries as training becomes slower. For example, extremely low dimensionality (e.g., 100) achieves an average performance of 60.29, while performance slightly declines as dimensionality increases from 500 to 2,000. We observe that a dimensionality of 500 strikes the best balance between query efficiency and performance, achieving an average score of 62.50, making it a practical choice for most scenarios.
> > > >
> > > > `Rank`
> > > > | Rank | Caltech | Pets | Flowers | Food | Aircraft | SUN | DTD  | SVHN | EuroSAT | Resisc | CLEVR | UCF | IN | Avg |
> > > > |----|----|----|----|----|----|----|----|----|----|----|----|----|----|----|
> > > > | 1 | 92.72 | 90.89 | 68.43 | 85.94 | 25.31 | 61.99 | 44.05 | **45.60** | 59.98 | 64.26 | 21.71 | 68.34 | **66.10** | 61.17 |
> > > > | 3 | 92.64 | **91.78** | 69.35 | **86.40** | 26.17 | 62.12 | 43.50 | 41.84 | 62.73 | 62.71 | **26.89** | 68.35 | 66.02 | 61.57 |
> > > > | 5 | **93.39** | 91.74 | **69.97** | 86.31 | **26.62** | **62.17** | **47.83** | 44.15 | **64.21** | **65.22** | 25.09 | **69.81** | 65.99 | **62.50** |
> > > >
> > > > Lower rank simplifies the model, improving query efficiency but potentially reducing expressiveness. As the rank increases, performance generally improves. In this experiment, rank 5 achieves the highest average score of 62.50, indicating it provides the best balance between efficiency and performance.

---

> > > > > ### Author Response · Authors · 2024-11-29
> > > > > **Awaiting your feedback**
> > > > >
> > > > > Dear reviewer Fqbz,
> > > > >
> > > > > We sincerely appreciate your thoughtful review and valuable feedback, which have been invaluable in improving our research. It has been a while since we posted our last response, so we would like to follow up to see whether our response has sufficiently addressed your concerns. If there is anything else the reviewer wants us to address more, please let us know. We would be happy to engage in any further discussion.
> > > > >
> > > > > Best regards,\
> > > > > Authors

---

> > > > > > ### Author Response · Authors · 2024-12-04
> > > > > > **closing**
> > > > > >
> > > > > > Dear Reviewer Fqbz
> > > > > >
> > > > > > Thank you for taking the time to provide thoughtful feedback on our manuscript. We have done our best to address your concern regarding ZIP, including clarifying the effects of feature sharing and explaining its generalization performance. We hope our responses have effectively clarified your concerns and provided the information you were looking for.
> > > > > >
> > > > > > Best regards, \
> > > > > > The Authors

---

### Official Review · Reviewer_Tcth · 2024-11-02

**Soundness:** 3
**Presentation:** 4
**Contribution:** 3
**Rating:** 6
**Confidence:** 3

**Summary:**

The paper proposes a method to optimize black-box models without the need for computing gradients (zeroth-order). The key observation is that increasing the number of learnable parameters in soft prompts hurts the performance and training speed of zeroth-order optimization, while this trend is reversed for SGD-based prompt tuning (first-order). To overcome this, authors propose to reparameterize soft prompts in order to reduce the effective number of learnable parameters while maintaining the extrinsic embedding dimensionality. The proposed reparameterization involves projecting parameters into a diagonal matrix, feature sharing and gradient clipping. In addition, reducing the number of learnable parameters results in increased query efficiency (reduced number of forward passes through the model). The proposed method is applied to black-box prompt-tuning of a CLIP model, and evaluated on a suite of standard vision-language benchmarks, achieving improvements of 6% in few-shot accuracy and 48% in query efficiency compared to the best performing existing methods.

**Strengths:**

* Good motivation to reduce the number of learnable parameters in ZO optimization (section 3) and clever idea to reduce the intrinsic dimensionality while maintaining the number of tokens (and the extrinsic dimensionality, which is a requirement from the model being optimized).
* Several techniques (diagonal matrix, parameter sharing) are applied to preserve performance while reducing the number of learnable parameters.
* The proposed method not only improves few-shot performance wrt existing ZO methods but also reduces considerably the number of function evaluations required to reach a certain level of performance (section 5.3).
* All the design choices for the soft prompt reparameterization are thoroughly ablated in section 6.
* The paper is clearly written and easy to follow.

**Weaknesses:**

* Authors motivate fine-tuning black-box models with the use case of improving proprietary LLMs (e.g. GPT-4, Gemini) which are only accessible through API. However, this interface only accepts text and images as input, not soft prompts or embeddings, so the proposed method would not be directly applicable to API-based models.
* To verify the method's robustness and generality, it should be evaluated on other model families such as multimodal LLMs.
* Figures 2, 4, 6 and 7a should report validation accuracy since there could be overfitting.

**Questions:**

* It is not until the background section that I understood what zeroth-order intrinsic-dimensional prompt-tuning means. I suggest to improve the introduction to make it clearer from early on.
* In figure 2, it would be good to add a baseline of accuracy when no soft prompts are optimized (i.e. m=0).
* Where are the learned soft prompts injected? Are they concatenated to text embeddings and fed to CLIP's text encoder?
* In table 3, the average accuracies for CDT between ZIP and the second-best method seem very close. Did authors run a significance test?

---

> ### Author Response · Authors · 2024-11-22
> **Response to Reviewer Tcth (part 1)**
>
> We sincerely thank the reviewer for acknowledging the contributions of our work and offering constructive feedback. We address the reviewer’s specific comments below and we would be keen to engage in any further discussion.
>
> ---
>
> **Soft prompts not directly applicable?**
> > Authors motivate fine-tuning black-box models with the use case of improving proprietary LLMs (e.g. GPT-4, Gemini) which are only accessible through API. However, this interface only accepts text and images as input, not soft prompts or embeddings, so the proposed method would not be directly applicable to API-based models.
>
> Thank you for this insightful comment. We acknowledge that our soft prompt-based approach cannot be directly applied to the scenario described by the reviewer. Our study builds on the Language-model-as-a-Service (LMaaS) scenario [1], widely recognized in prior works as a plausible framework for black-box prompt tuning [2, 3]. While GPT-4 or Gemini do not currently support this specific scenario, we believe that exploring the LMaaS approach is essential for enabling more flexible and efficient fine-tuning of black-box models via APIs.
>
> ---
>
> **Apply to other model families?**
> > It should be evaluated on other model families such as multimodal LLMs for robustness and generality.
>
> Thank you for the suggestion. It would be interesting to see how ZIP evaluates on multimodal foundation models such as LLaVA to demonstrate its robustness and generality. Unfortunately, due to resource and time limitations during the rebuttal period, we were unable to conduct such experiments.
>
> Instead, we provide additional results on SigLIP [4], another vision-language model distinct from CLIP, in the table below. While ZIP does not perform the best for all datasets, it achieves significantly higher accuracy on several datasets such as DTD, SVHN, EuroSAT, Resisc45, CLEVR, and UCF101, and records the highest on average, demonstrating its robustness and generality of our method compared to existing other BBPT methods.
>
> | Method | Caltech | Pets | Flowers | Food | Aircraft | SUN | DTD  | SVHN | EuroSAT | Resisc | CLEVR | UCF | IN | Avg |
> |----|----|----|----|----|----|----|----|----|----|----|----|----|----|----|
> | BAR | **85.8** | **82.2** | **68.0** | **58.3** | 10.7 | 23.3 | 47.2 | 15.2 | 39.6 | 23.6 | 26.1 | 23.3 | **54.9** | 42.9 |
> | BLackVIP | 81.6 | 78.9 | 56.6 | 47.8 | 9.6 | **46.3** | 41.1 | 30.7 | 22.5 | 37.0 | 26.3 | 37.0 | 49.4 | 43.4 |
> | BPTVLM | 63.9 | 71.7 | 47.7 | 37.2 | 9.9 | 35.5 | 45.6 | 14.7 | 44.8 | 37.0 | 24.8 | 35.5 | 39.5 | 39.1 |
> | ZIP | 72.1 | 82.0 | 59.3 | 40.9 | **11.8** | 45.3 | **49.4** | **34.2** | **48.7** | **44.3** | **31.4** | **40.6** | 49.7 | **46.9** |
>
> ---
>
> **Need validation accuracy**
> > Figures 2, 4, 6 and 7a should report validation accuracy since there could be overfitting.
>
> Thank you for pointing this out. To address concerns about overfitting, we now include validation accuracy for all cases in Figure 9, 10, 11, and 12 in Appendix A. The results show that training accuracy and validation accuracy generally exhibit similar trends, with no significant signs of overfitting observed.
>
> ---
>
> **Clarify in Introduction**
> > It is not until the background section that I understood what zeroth-order intrinsic-dimensional prompt-tuning means. I suggest to improve the introduction to make it clearer from early on.
>
> We appreciate this feedback. We will make sure to revise the paper and clarify what  “zeroth-order intrinsic-dimensional prompt-tuning” means earlier in Introduction.
>
> &nbsp;
>
> [1] Sun, Tianxiang, et al. “Black-box tuning for language model as-a-service.” ICML, 2022.\
> [2] Yu, Lang, et al. “Black-box prompt tuning for vision-language model as-a-service.” IJCAI, 2023.\
> [3] Song, Jiang-Long, et al. “Competition solution for prompt tuning using pretrained language model.” arXiv preprint arXiv:2212.06369 (2022).\
> [4] Zhai, Xiaohua, et al. “Sigmoid loss for language image pre-training.” ICCV, 2023.

---

> > ### Author Response · Authors · 2024-11-22
> > **Response to Reviewer Tcth (part 2)**
> >
> > **Adding baseline for Figure 2**
> > > In figure 2, it would be good to add a baseline of accuracy when no soft prompts are optimized (i.e. m=0).
> >
> > Thank you for the suggestion. In response, we have updated Figure 9 to include a baseline accuracy where no soft prompts are optimized (i.e., manual prompt) for better comparison.
> >
> > ---
> >
> > **Injection of learned prompts**
> > > Where are the learned soft prompts injected? Are they concatenated to text embeddings and fed to CLIP's text encoder?
> >
> > Thank you for the question. The learned soft prompts are prepended to the text embeddings, as in CoOp, and subsequently fed into the text encoder of CLIP.
> >
> > ---
> >
> > **Significance test for Table 3**
> > > In table 3, the average accuracies for CDT between ZIP and the second-best method seem very close. Did authors run a significance test?
> >
> > Thank you for raising this point. To address the concern, we have included standard deviations in Table 3 to better reflect statistical significance.
> >
> > `Cross-Dataset Transfer`
> > | Method | IN | Caltech | Pets | Flowers | Food | Aircraft | SUN | DTD  | SVHN | EuroSAT | Resisc | CLEVR | UCF | Avg |
> > |----|----|----|----|----|----|----|----|----|----|----|----|----|----|----|
> > | BAR | 64.0 (0.1) | 92.3 (0.2) | 84.3 (0.1) | 64.3 (0.1) | 83.1 (0.1) | 20.8 (0.1) | 61.0 (0.0) | 42.2 (0.2) | 20.0 (0.9) | **49.6** (0.6) | 50.6 (0.5) | 14.5 (0.1) | 63.0 (0.2) | 53.8 |
> > | BlackVIP | 65.5 (0.5) | **92.5** (0.3) | **86.2** (1.3) | 64.9 (0.4) | **83.6** (0.4) | **22.3** (0.4) | **62.0** (0.3) | **43.3** (1.0) | 18.7 (2.0) | 40.5 (1.1) | **55.7** (1.1) | **15.2** (1.0) | **64.1** (0.2) | 54.1 |
> > | BPTVLM | 55.5 (1.2) | 80.7 (1.4) | 77.7 (2.2) | 50.3 (8.6) | 77.6 (0.4) | 16.3 (2.0) | 43.8 (5.4) | 30.8 (0.8) | 15.5 (4.1) | 34.6 (9.6) | 37.7 (8.5) | 12.4 (1.4) | 54.8 (1.3) | 44.4 |
> > | ZIP | **66.0** (1.2) | 90.4 (1.8) | 85.6 (1.5) | **65.6** (1.3) | **83.6** (1.3) | 20.5 (0.1) | 60.6 (1.6) | 40.9 (2.6) | **27.0** (1.0) | 42.3 (3.4) | 55.6 (2.0) | 14.5 (1.3) | 63.6 (1.1) | **54.2** |
> >
> > `Out-of-Distribution Generalization`
> > | Method | ImageNet-A | ImageNetV2 | ImageNet-R | ImageNet-Sketch | Average |
> > |----|----|----|----|----|----|
> > | BAR | 40.2 (0.1) | 57.5 (0.1) | 72.0 (0.0) | 43.8 (0.1) | 53.4 |
> > | BlackVIP | 42.5 (1.7) | 59.2 (0.7) | 73.1 (0.5) | 44.6 (0.4) | 54.9 |
> > | BPTVLM | 32.7 (1.2) | 46.7 (3.2) | 61.7 (4.5) | 33.5 (1.9) | 43.7 |
> > | ZIP | **47.8** (0.7) | **59.5** (1.5) | **74.7** (0.9) | **45.4** (1.5) | **56.9** |

---

> > > ### Comment · Reviewer_Tcth · 2024-11-26
> > >
> > > Thank you for your thorough response.
> > >
> > > Could you please comment on the performance of ZOO compared to m=0 (figure 9)? It seems that, for several datasets (e.g., Flowers102, Food101, FGVCAircraft, UCF101), optimizing soft prompts with ZOO actually hurts performance.
> > >
> > > Could you also assess the statistical significance of the reported results based on the standard deviations reported for table 3? For several datasets where ZIP achieves the highest accuracy (e.g., IN, Flowers102, Food101, ImageNetV2, ImageNet-Sketch), it seems there's overlap with the std of the second-best result (usually BlackVIP).

---

> > > > ### Author Response · Authors · 2024-11-28
> > > > **Response to the official comment by Reviewer Tcth**
> > > >
> > > > Thank you for carefully reviewing our response. We address the reviewer’s additional comments below.
> > > >
> > > > ---
> > > >
> > > > > Could you please comment on the performance of ZOO compared to m=0 (figure 9)? It seems that, for several datasets (e.g., Flowers102, Food101, FGVCAircraft, UCF101), optimizing soft prompts with ZOO actually hurts performance.
> > > >
> > > > First of all, we would like to clarify that the result we presented as m=0 in Figure 9 was actually “manual prompting” rather than “no prompts” (as mentioned as “manual prompt (m=0)” in L809). However, we realize now that what the reviewer has asked us to include as a baseline is literally “no prompts” (i.e., without any prompting at all). We have updated the paper to include the results for “no prompts”. Please let us know if we need to address the reviewer’s request further.
> > > >
> > > >
> > > >
> > > >
> > > > ---
> > > >
> > > > > Could you also assess the statistical significance of the reported results based on the standard deviations reported for table 3? For several datasets where ZIP achieves the highest accuracy (e.g., IN, Flowers102, Food101, ImageNetV2, ImageNet-Sketch), it seems there's overlap with the std of the second-best result (usually BlackVIP).
> > > >
> > > > To provide more statistically robust results, we conducted experiments using an extended set of 10 seeds (1–10). Based on this, we performed a t-test to compare p-values for the significance test. The results showed that ZIP demonstrated statistically significant superior performance in OOD, whereas ZIP and BlackVIP exhibited comparable performance in CDT. Please refer to the table below for detailed results and analysis.
> > > >
> > > > `Cross Dataset Transfer`
> > > > | Method | IN | Caltech | Pets | Flowers | Food | Aircraft | SUN | DTD  | SVHN | EuroSAT | Resisc | CLEVR | UCF | Avg |
> > > > |----|----|----|----|----|----|----|----|----|----|----|----|----|----|----|
> > > > | BlackVIP | 65.19 | 92.69 | 86.28 | 64.72 | 83.36 | 22.31 | 62.04 | 42.88 | 17.68 | 39.71 | 55.98 | 15.70 | 64.07 | 54.82 |
> > > > | ZIP | 65.91 | 91.69 | 85.74 | 65.64 | 84.54 | 20.67 | 60.10 | 39.39 | 21.42 | 44.43 | 54.52 | 14.39 | 61.99 | 54.65 |
> > > > | t-stats | 1.745 | -1.879 | -0.690 | 2.942 | 2.974 | -8.771 | -2.755 | -4.349 | 2.615 | 3.409 | -1.588 | -2.479 | -2.650 | |
> > > > | p-value | 0.098 | 0.076 | 0.498 | 0.009 | 0.008 | 6.45e-08 | 0.013 | 3.87e-04 | 0.018 | 0.003 | 0.130 | 0.023 | 0.016 | |
> > > >
> > > > In CDT tasks, significance tests indicate that ZIP performs significantly better on Flowers (p=0.009), Food (p=0.008), SVHN (p=0.018), and EuroSAT (p=0.003). In contrast, BlackVIP achieves significantly higher performance on Aircraft (p=6.45e-08), DTD (p=3.87e-04), CLEVR (p=0.023), and UCF (p=0.016) (p < 0.05).
> > > >
> > > > `Out-of-Distribution Generalization`
> > > > | Method | ImageNet-A | ImageNetV2 | ImageNet-R | ImageNet-Sketch | Average |
> > > > |----|----|----|----|----|----|
> > > > | BlackVIP | 41.90 | 58.85 | 72.81 | 44.32 | 54.47 |
> > > > | ZIP | 47.94 | 59.48 | 74.64 | 45.25 | 56.82 |
> > > > | t-stats | 10.299 | 1.549 | 4.359 | 2.619 | |
> > > > | p-value | 5.66e-09 | 0.139 | 3.78e-4 | 0.017 | |
> > > >
> > > > In OOD tasks, ZIP consistently outperforms BlackVIP in average performance (ZIP: 56.82, BlackVIP: 54.47). Notably, ZIP shows statistically significant improvements on ImageNet-A (p=5.66e-09), ImageNet-R (p=3.78e-4), ImageNet-Sketch (p=0.017) while the performance difference on ImageNetV2 (p=0.139) is smaller and not statistically significant (p > 0.05).
> > > >
> > > > BlackVIP’s strength in generalization performance stems from its design and objectives. It is specifically tailored to enhance generalization through an image-dependent prompting strategy, drawing inspiration from prior work [1]. In contrast, ZIP prioritizes query efficiency, which differentiates it from BlackVIP in terms of its primary goal. Despite these different focuses, ZIP outperforms BlackVIP in OOD and base-to-new generalization tasks while still delivering competitive performance in CDT.
> > > >
> > > > We sincerely appreciate the reviewer for the opportunity to improve the reliability of our results. If there are any additional points to discuss, please let us know.
> > > >
> > > > &nbsp;
> > > >
> > > > [1] Zhou, Kaiyang, et al. "Conditional prompt learning for vision-language models." CVPR, 2022.

---

> > > > > ### Comment · Reviewer_Tcth · 2024-12-03
> > > > >
> > > > > Thanks for your follow-up!
> > > > >
> > > > > I am not sure I understand the concept of "no prompt", as CLIP always computes similarity between an image and a text prompt. Does it mean that you used the same prompts as in the original paper, without further prompt engineering? In any case, now in figure 9 it seems that ZOO with m=1 consistently outperforms m=0, which is reassuring as I'd expect any optimization method to perform better than the baseline (m=0 in this case).
> > > > >
> > > > > From the additional significance results for table 3, I understand that on CDT, ZIP and BlackVIP have comparable performance and the advantage of ZIP lies in its query efficiency. I assume figure 10 shows query efficiency for the few-shot setting (table 1). Do these curves look similar for the setting corresponding to table 3?

---

> > > > > > ### Author Response · Authors · 2024-12-04
> > > > > > **Response to the official comment by Reviewer Tcth**
> > > > > >
> > > > > > > I am not sure I understand the concept of "no prompt", as CLIP always computes similarity between an image and a text prompt. Does it mean that you used the same prompts as in the original paper, without further prompt engineering? In any case, now in figure 9 it seems that ZOO with m=1 consistently outperforms m=0, which is reassuring as I'd expect any optimization method to perform better than the baseline (m=0 in this case).
> > > > > >
> > > > > > Yes, the term "no prompt" means there are no prompt tokens that are further engineered. Specifically, we only used the [CLASS] token for $m=0$ case, without any additional soft prompts. Since we append soft prompt tokens in front of [CLASS] token (i.e. “X X X [CLASS]”, where X are soft prompt tokens), we decided to use only [CLASS] token for $m=0$, to provide a fair baseline with respect to m. Sorry for the confusion, we will revise the naming of $m=0$ from “no prompt” to “no engineered prompt” for the final version.
> > > > > >
> > > > > >
> > > > > >
> > > > > > ---
> > > > > >
> > > > > > > From the additional significance results for table 3, I understand that on CDT, ZIP and BlackVIP have comparable performance and the advantage of ZIP lies in its query efficiency. I assume figure 10 shows query efficiency for the few-shot setting (table 1). Do these curves look similar for the setting corresponding to table 3?
> > > > > >
> > > > > > To evaluate on CDT and OOD tasks, the prompt is first trained on ImageNet (same one in Table 1), and then the resulting trained model is tested on CDT and OOD tasks **without any parameter tuning**. As a result, there would be no dedicated training curves specific to the CDT and OOD tasks.
> > > > > >
> > > > > > To analyze query efficiency for CDT and OOD, one potentially indirect way would be to increase the available budget during ImageNet training and then evaluate the performance on CDT and OOD tasks. This would provide valuable insights into how query efficiency impacts task performance. We greatly appreciate the reviewer’s thoughtful suggestion, which has inspired this idea, and we plan to include it in the final version of the paper.
> > > > > >
> > > > > > ---
> > > > > >
> > > > > > We sincerely thank the reviewer for the constructive feedback and insightful discussion. These exchanges have allowed us to conduct a more comprehensive analysis of our method and further solidify its validation.

---

### Official Review · Reviewer_BGp2 · 2024-11-05

**Soundness:** 2
**Presentation:** 3
**Contribution:** 3
**Rating:** 5
**Confidence:** 2

**Summary:**

The paper introduces ZIP for efficient zeroth-order prompt-tuning of black-box vision-language models. ZIP addresses the challenge of excessive query requirements in existing black-box prompt-tuning methods by reducing problem dimensionality and gradient estimate variance through feature sharing and intrinsic-dimensional gradient clipping. ZIP demonstrates significant improvements in few-shot accuracy and query efficiency over other existing methods. Various experiments on image classification show the effectiveness of ZIP.

**Strengths:**

- ZIP is well-motivated.
- The paper is well-organized.
- Empirical analyses of the proposed method are sufficient.

**Weaknesses:**

I'm not familiar with this research field, i.e. black box prompt tuning. Therefore, it's hard for me to accurately judge the novelty of the proposed method compared with existing works.

From my perspective, one major weakness is that I find the competitors in the experiments are slightly old, e.g. BLACKVIP is published at CVPR'23 and BPTVLM is published at IJCAI'23. There are some more recent works like [a][b] in this field. I think the authors should better discuss the differences between ZIP and more recent works like [a][b], and provide fair experimental comparisons as well.

[a] Language Models as Black-Box Optimizers for Vision-Language Models, CVPR 2024, https://openaccess.thecvf.com/content/CVPR2024/html/Liu_Language_Models_as_Black-Box_Optimizers_for_Vision-Language_Models_CVPR_2024_paper.html
[b] Connecting the Dots: Collaborative Fine-tuning for
Black-Box Vision-Language Models, ICML 2024, https://arxiv.org/abs/2402.04050

**Questions:**

See weaknesses.

---

> ### Author Response · Authors · 2024-11-22
> **Response to Reviewer BGp2 (part 1)**
>
> We sincerely thank the reviewer for acknowledging the clear motivation of our method and providing constructive feedback. We address the reviewer’s specific comments below and welcome any further discussion.
>
> ---
>
> **Novelty?**
> > I'm not familiar with this research field, i.e. black box prompt tuning. Therefore, it's hard for me to accurately judge the novelty of the proposed method compared with existing works.
>
> Thank you for the comment. We will first outline the problem context and then explain our proposed solutions.
>
> Black-box models that are accessible only through APIs can be trained using black-box prompt tuning (BBPT). However, users are typically constrained by limited API budgets, making **query efficiency** a critical factor for the practicality of BBPT. Despite this, prior works have not adequately addressed this issue. For instance, BlackVIP and BPTVLM often require tens of thousands of queries, which severely limits their practicality. To address this, we are the first to explicitly highlight this problem and propose several techniques to significantly improve query efficiency.
>
> Specifically, we introduce two key technical contributions:
> * To address the challenge of **dimensionality dependency** (i.e., the number of queries scales with the dimensionality of the problem), we propose a novel low-rank representation. This approach reduces the dimensionality while effectively mitigating the loss of expressive power through feature sharing.
> * High variance in zeroth-order information can significantly degrade query efficiency. To tackle this, we propose a threshold-free gradient clipping method, termed “intrinsic dimensional clipping”. Inspired by prior studies on clipping thresholds [1, 2, 3], we set the clipping threshold to $\sqrt{d}$, which corresponds to the standard deviation of the zeroth-order gradient, where $d$ is the dimensionality of the problem. This approach not only reduces the variance of zeroth-order information but also achieves near-optimal performance without requiring manual tuning (See Figure 7b and 20 in revisioned paper).
>
> &nbsp;
>
> [1] Zhang, Bohang, et al. “Improved analysis of clipping algorithms for non-convex optimization.” NeurIPS, 2020.\
> [2] Zhang, Jingzhao, et al. “Why are adaptive methods good for attention models?” NeurIPS, 2020.\
> [3] Zhang, Jingzhao, et al. “Why gradient clipping accelerates training: A theoretical justification for adaptivity.” arXiv preprint arXiv:1905.11881 (2019).

---

> > ### Author Response · Authors · 2024-11-22
> > **Response to Reviewer BGp2 (part 2)**
> >
> > **Compare to more recent works [1, 2]**
> > > From my perspective, one major weakness is that I find the competitors in the experiments are slightly old, e.g. BLACKVIP is published at CVPR'23 and BPTVLM is published at IJCAI'23. There are some more recent works like [1, 2] in this field. I think the authors should better discuss the differences between ZIP and more recent works like [1, 2], and provide fair experimental comparisons as well.
> >
> > We understand the reviewer’s concern regarding the lack of comparisons with recent works [1, 2]. To address this, we first describe the differences between [1], [2], and our method in the table below. Additionally, we highlight that both methods are based on assumptions different from ours, making a strict comparison challenging.
> > | Method | Prompt | Optimizer | Access Permission |
> > |----|----|----|----|
> > | [1] | Hard | LLM (ChatGPT) | Loss |
> > | [2] | Soft | CMA-ES | Logits |
> > | ZIP | Soft | ZOO | Loss |
> > * First, [1] proposes a method to enable user-specific utilization of black-box VLMs by learning hard prompts through an LLM. This approach uses an LLM as an optimizer to fine-tune the VLM, and separate APIs are required for both the LLM and VLM. This makes direct comparisons with ZIP difficult as our approach operates under a single API setting.
> > * Similarly, [2] proposes a method to learn soft prompts for black-box VLMs using CMA-ES, an evolutionary algorithm. This approach assumes access to the logits of VLMs, which is a relatively relaxed black-box scenario. In contrast, our approach relies solely on the single loss value of the VLM, making direct comparisons infeasible.
> >
> > We emphasize that our method achieves state-of-the-art results under the widely adopted setting following prior works, such as BlackVIP and BPTVLM (i.e., using soft prompts as input and receiving a single loss value as output).
> >
> > ---
> >
> > Thank you for the constructive comments and thoughtful suggestions. We appreciate the opportunity to address your concerns and clarify the key aspects of our work. We have provided detailed responses to each of your points, including the novelty of our method and comparisons with recent works. We hope these clarifications address your concerns.
> >
> > If there are any additional points that you believe we should address, please let us know. Otherwise, we would be sincerely grateful if the reviewer could reconsider the overall rating in light of our responses.
> >
> > &nbsp;
> >
> > [1] Liu, Shihong, et al. “Language models as black-box optimizers for vision-language models.” CVPR, 2024.\
> > [2] Wang, Zhengbo, et al. “Connecting the dots: Collaborative fine-tuning for black-box vision-language models.” ICML, 2024.

---

> ### Author Response · Authors · 2024-11-28
> **Dear Reviewer**
>
> Dear reviewer BGp2
>
> We sincerely appreciate your valuable feedback and constructive comments. Given that some time has passed since we shared our response, we kindly ask whether it has adequately addressed your concerns. Please let us know if there is anything else we need to address. We would be happy to discuss further. Otherwise, if you find our response reasonably satisfactory, we would greatly appreciate it if you could consider re-evaluating your initial rating.
>
> Best wishes, \
> The authors.

---

> ### Author Response · Authors · 2024-12-02
> **Awaiting your response**
>
> Dear reviewer BGp2
>
> As the discussion period is closing soon, we would like to see if our response has sufficiently addressed your concerns. Should there be any remaining issues, please do not hesitate to let us know. If you find our responses satisfactory, we would be sincerely grateful if you could consider revisiting your initial rating.
>
> Best wishes,\
> The authors.

---

### Author Response · Authors · 2024-12-02
**Global Response**

Dear reviewers (`BGp2`, `Tcth`, `Fqbz`, `NLRG`) and area chair,

We sincerely appreciate the time and effort you dedicated to reviewing our paper. Your insightful feedback has been invaluable in refining our research.

We are pleased that most reviewers recognized the key strengths of our work. During the discussion period, we also provided clarifications requested by reviewers (**need validation accuracy** (`Tcth`), **adding baseline for Figure 2** (`Tcth`), **explanation of generalization** (`Fqbz`)). To further support our approach, we conducted additional ablation studies (**apply to other model families** (`Tcth`), **ablation for different combinations of modules** (`NLRG`), **feature sharing and robustness** (`Fqbz`)).

Below, we summarize the **Strengths highlighted by reviewers $\color{blue}{[S]}$**, and **Key contributions of our work $\color{cyan}{[C]}$**.

---

### **Strengths highlighted by reviewers $\color{blue}{[S]}$**
#### **Novelty**
- “innovatively enhances zeroth-order prompt tuning through intrinsic-dimensional gradient clipping and low-rank parameterization” (`Fqbz`)
- “clever idea to reduce the intrinsic dimensionality while maintaining the number of tokens” (`Tcth`)
- “a novel black-box prompt-tuning method” (`NLRG`)

#### **Motivation**
- “good motivation to reduce the number of learnable parameters in ZO optimization” (`Tcth`)
- “ZIP is well-motivated” (`BGp2`)

#### **Evaluations**
- “comprehensive evaluations demonstrate ZIP's superior accuracy and query efficiency across 13+ tasks” (`Fqbz`)
- “improves few-shot performance wrt existing ZO methods” (`Tcth`)
- “the paper demonstrates strong performance across three extensive and diverse experimental settings” (`NLRG`)
- “all the design choices for the soft prompt reparameterization are thoroughly ablated in section 6.” (`Tcth`)
- “the ablation studies further support the approach, particularly highlighting that the feature-sharing technique helps preserve the model’s expressive capacity.” (`NLRG`)

#### **Practicality**
- “the proposed approach is well-suited for practical applications with limited query budgets” (`NLRG`)
- “proving its practical value under query constraints” (`Fqbz`)
- “the intrinsic-dimensional clipping mechanism in ZIP requires no manual hyperparameter tuning, making it highly practical and user-friendly.” (`NLRG`)

---

### **Key contributions of our work $\color{cyan}{[C]}$**
#### **Problem identification**
Black-box prompt tuning (BBPT), a method for training black-box models, is inherently restricted by limited API budgets, making query efficiency a crucial factor for its practicality. Despite this, prior works have not addressed this issue. For instance, existing works often require tens of thousands of queries, which severely limits their practicality. We are the first to explicitly identify this problem in the language-model-as-a-service scenario and propose a novel idea to significantly improve query efficiency.

&nbsp;

#### **Technical contributions**
We introduce two key technical contributions to tackle these challenges:
1. **Low-rank representation for dimensionality dependency**: To address the challenge of dimensionality dependency of zeroth-order methods (i.e., the number of queries scales with the dimensionality of the problem), we propose a novel low-rank representation. This approach reduces the dimensionality while effectively mitigating the loss of expressive power through feature sharing.

2. **Intrinsic dimensional clipping for variance reduction**: High variance in zeroth-order information can significantly degrade query efficiency. To tackle this, we propose a threshold-free gradient clipping method, termed “intrinsic dimensional clipping”. The clipping threshold is set to $\sqrt{d}$, which corresponds to the standard deviation of the zeroth-order gradient, where $d$ is the dimensionality of the problem. This approach not only reduces the variance of zeroth-order information but also achieves near-optimal performance without requiring manual tuning.

&nbsp;

####  **Experimental results**
Our method achieves state-of-the-art performance in few-shot learning, base-to-new generalization, and out-of-distribution tasks under limited query budgets, while also delivering competitive results in cross-dataset transfer. Furthermore, it reduces the number of queries required to achieve a specific accuracy level by 48% compared to the best-performing alternative BBPT methods in few-shot accuracy. These results highlight the effectiveness of our approach in tackling key challenges in BBPT.

---

### Meta-Review · Area_Chair_o6PK · 2024-12-17

**Metareview:**

The paper tackles an important problem in black-box prompt tuning, i.e., existing methods rely on excessive queries for model update. The solutions proposed in this work include a low-rank reparameterization method for reducing learnable parameters and a gradient-clipping method for reducing the variance of zeroth-order gradients. The paper received four reviews with 3x borderline accept and 1x borderline reject. However, the "negative" reviewer indicated that he/she is not familiar with the topic and therefore not confident in his/her decision (the AC discounts this review). The rest of the reviewers are generally positive about this work: they found the paper well-motivated, the idea novel, and the performance strong.

**Additional Comments On Reviewer Discussion:**

The major concerns raised by the reviewers are about lack of comparisons with some baselines and lack of ablation studies on the combination of different modules proposed in this work. The authors have provided a comprehensive rebuttal to address these issues. The AC has read the rebuttal and conversations between the authors and the reviewers and found them satisfactory. The AC strongly suggests that the authors add the additional ablation results and the results of applying their method to different model families to the camera ready.

---

### Decision · Program_Chairs · 2025-01-22

Accept (Poster)